# LEARNING TO CLARIFY:
# MULTI-TURN CONVERSATIONS WITH ACTION-BASED CONTRASTIVE SELF-TRAINING

**Maximillian Chen**[*1,2]**, Ruoxi Sun**[1]**, Tomas Pfister**[1]**, Sercan Ö. Arık**[1]
[1]Google [2]Columbia University
maxchen@cs.columbia.edu, {ruoxis,tpfister,soarik}@google.com

## ABSTRACT

Large language models (LLMs), optimized through human feedback, have rapidly emerged as a leading paradigm for developing intelligent conversational assistants. However, despite their strong performance across many benchmarks, LLM-based agents might still lack conversational skills such as disambiguation – when they are faced with ambiguity, they often overhedge or implicitly guess users' true intents rather than asking clarification questions. Under task-specific settings, high-quality conversation samples are often limited, constituting a bottleneck for LLMs' ability to learn optimal dialogue action policies. We propose Action-Based Contrastive Self-Training (*ACT*), a quasi-online preference optimization algorithm based on Direct Preference Optimization (DPO), that enables data-efficient dialogue policy learning in multi-turn conversation modeling. We demonstrate *ACT*'s efficacy under in data-efficient tuning scenarios, even when there is no action label available, using multiple real-world conversational tasks: tabular-grounded question-answering, machine reading comprehension, and AmbigSQL, a novel task for disambiguating information-seeking requests for complex SQL generation towards data analysis agents. Additionally, we propose evaluating LLMs' ability to function as conversational agents by examining whether they can implicitly recognize and reason about ambiguity in conversation. *ACT* demonstrates substantial conversation modeling improvements over standard tuning approaches like supervised fine-tuning and DPO.

## 1 INTRODUCTION

Conversations offer a natural and effective way for humans and intelligent systems to collaborate (Amershi et al., 2019; Lemon, 2012). The impressive capabilities of large language models (LLMs) have powered the rapid development of many generalist conversational assistants such as ChatGPT[1] and Gemini (Gemini Team et al., 2023), which present an opportunity for users to verbalize their need for assistance on complex tasks. However, the promises of a conversational interfaces also come with the complexities of language. Human conversation is riddled with ambiguity, whether it be due to humans' tendency to underspecify (Zipf, 1949) or even due to syntactic errors (Messer, 1980). Moreover, disambiguation becomes even more important in complex domains where it can be a difficult multi-turn process to achieve common ground (Beers et al., 2006). As it stands, existing LLM-powered conversational agents continue to struggle with modeling ambiguity (Liu et al., 2023), and tend to exhibit unwanted behavior such as overhedging (Ouyang et al., 2022) or generating responses which represent a "guess" of the user's intent (Deng et al., 2023a) (see Figure 1).

One of the primary reasons that LLMs may exhibit unwanted conversational behaviors is that their language modeling objective during pre-training or supervised fine-tuning (SFT) is not directly aligned with this goal  (Ouyang et al., 2022). While approaches like Ouyang et al. (2022) propose LLM "alignment" using post-training approaches like reinforcement learning from human feedback (RLHF) (Christiano et al., 2017), existing models still struggle with conversational tasks spanning

---

* Work done during an internship at Google.
[1]https://openai.com/blog/chatgpt

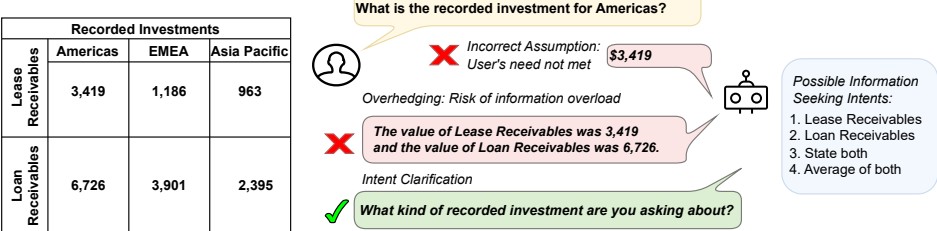

Figure 1: Simplified example of ambiguity present at tabular-grounded conversational question answering based on Deng et al. (2022). A conversational agent should recognize when there is ambiguity and ask a clarifying question towards a more accurate final answer.

multiple turns (Wang et al., 2023). This is partly due to the fact that existing approaches do not directly optimize for pragmatic skills (e.g. Bai et al. (2022)). Moreover, there is often high variance in the target distribution of a particular use case, so it is imperative that downstream adaptation approaches can effectively steer LLM policies. Given large-scale in-distribution training data, this may be feasible with standard SFT or RLHF. But, dialogue policy learning can be particularly challenging given limited data (Chen et al., 2022a; Dong et al., 2023) and collecting high-quality conversational datasets can be difficult for reasons such as annotation costs and privacy concerns (Chen et al., 2023a).

This motivates the design of a conversational adaptation approach for LLMs which is more closely aligned with the goal of modeling actions in multi-turn conversation. We focus on improving LLMs' abilities to implicitly select conversational strategies in ambiguous contexts, and propose an approach called Action-Based Contrastive Self-Training (*ACT*). *ACT* is a sample-efficient, quasi-online Direct Preference Optimization algorithm (Rafailov et al., 2024) which focuses on contrasting the differences between an agent's possible pragmatic conversational actions. We demonstrate *ACT*'s sample-efficient performance on a diverse range of mixed-initiative conversational tasks: (i) tabular-grounded question answering, (ii) machine reading comprehension, and (iii) text-to-

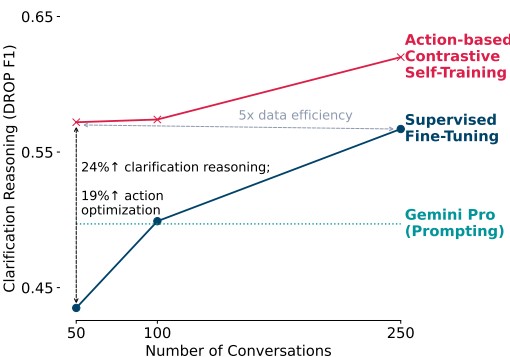

Figure 2: *ACT* greatly outperforms standard tuning approaches in data-efficient settings for conversational modeling, as exemplified here on PACIFIC.

SQL generation, demonstrating large improvements compared to standard adaptation approaches (see Figure 2). Our work highlights the necessity of considering action-based preferences for conversational tasks, and we propose a workflow for evaluating LLMs' ability to recognize and reason about ambiguity in conversation.

## 2 RELATED WORK

### 2.1 MIXED-INITIATIVE CONVERSATIONAL AGENTS

Neural approaches to building mixed-initiative conversational agents typically consist of two core components: an understanding and planning module (e.g., a binary prediction task to determine whether to ask a clarifying question or provide an answer), and a generation module which can be controlled at a pragmatic level using the output of the planning module (Chen et al., 2017; 2022b; Qian et al., 2022; Yu, 2017) (e.g., forming an utterance which follows the predicted action).

**Generation** Many existing works focus on novel training methodologies to improve conditional generation as a complement to planning, with approaches such as multi-objective SFT (Chen et al., 2022b; Wen et al., 2016) or introducing specialized embeddings for control codes (Keskar et al., 2019). LLMs have vastly improved performance in pragmatically-controlled generation (Chen et al., 2023b), but all of these approaches still depend on conversational planning. Planning remains a difficult task – natural interaction is not deterministic and often requires long-horizon planning.

**Planning** The planning task can be viewed as a stochastic Markov Decision Process (Wang et al., 2020b; Yang et al., 2021; Yu et al., 2023) in which some dialogue state is drawn from a potentially unknown distribution, given the previous dialogue state and an imposed action. However, the action itself is not literally presented to the interacting parties; rather, an action is a low-dimensional representation of the pragmatic intent carried by a given dialogue utterance (i.e., a dialogue act Sadek (1991); Stolcke et al. (2000); Wu et al. (2023); Yu & Yu (2021)). As such, training planning modules often requires complex long-horizon reasoning and simulation to model the responses and intents of each interacting party. Such efforts have examined combining neural models with search algorithms (Cheng et al., 2022; Väth et al., 2023; Yu et al., 2023) and simulation (Deng et al., 2023c; Wang et al., 2020a; Yu et al., 2023). However, such modular approaches can incur high computational overhead (Yu et al., 2023) and may result in error propagation while not directly optimizing for response quality itself. We propose directly optimizing dialogue action planning as an implicit subtask of response generation in mixed-initiative conversation contexts, as we discuss in Sec. 3.2.

## 2.2 LEARNING FOR LLM ALIGNMENT

The current paradigm of LLM training for downstream use cases consists of three phases: pre-training, supervised fine-tuning (SFT) for instruction-following, and tuning for alignment with human preferences (Tunstall et al., 2023; Rafailov et al., 2024; Lee et al., 2023; Ouyang et al., 2022). We primarily focus on the phase of tuning for alignment. These approaches typically start with an initial policy model obtained by conducting SFT on a target task ($\pi_{SFT}$), before performing tuning (often with RL), using contrastive preference examples (most commonly collected through human feedback (Ouyang et al., 2022) or a similar proxy like LLM-generated feedback (Lee et al., 2023)). In the case of online algorithms like PPO, a reward model is first fit over the preference examples so that it could be used for RL optimization (Ouyang et al., 2022). Such algorithms have certain advantages which may benefit the Markov Decision Process-like nature of conversations — namely, a diverse search space as opposed to a fixed dataset, flexible reward functions, and broader policy exploration. However, PPO is notoriously difficult to tune, and offline algorithms such as DPO (Rafailov et al., 2024), SLiC (Zhao et al., 2023), and IPO (Azar et al., 2024) have become widely adopted as an LLM adaptation approach because they bypasses explicit reward modeling and thus only require one set of hyperparameters to optimize (Huang et al., 2024; Rafailov et al., 2024; Zhao et al., 2023; Zheng et al., 2023) while still achieving similar empirical results given a fixed preference dataset.

**On-Policy DPO** Many of our contemporaries also question the limits of fully offline preference learning algorithms and have examined "online" variants of them (Guo et al., 2024; Xu et al., 2023; 2024b). Yuan et al. (2024) proposes iterative DPO, and Chen et al. (2024) proposes a variant where ground-truth responses are considered winning, and responses sampled from the previous iteration of a policy model are considered losing. Pang et al. (2024) applies a variant of iterative DPO to optimize externalized reasoning chains. Our work differs from these in that we are proposing a novel approach to customize LLMs for specific conversational settings, in particular, multi-turn conversational settings. While other works look at applying DPO to conversations in general (e.g. Sun et al. (2024)), their focus is still on single-turn response optimization. *ACT* considers multi-turn trajectories for preference optimization, and to our knowledge, our work is the first paper to consider contrastive learning on the basis of conversational actions.

## 3 METHODS

### 3.1 PROBLEM SETUP

We consider the task of tuning an LLM to function as a *mixed-initiative conversational agent*. Through a series of dialogue interactions with a user, the LLM is expected to assist the user by eventually providing a correct response to their request. Unlike the common agent interaction setting where users completely control the flow of interaction with the expectation that agents may autonomously complete tasks such as online shopping (Liu et al., 2024), mixed-initiative agents should understand how to redirect the flow of the interaction (Allen et al., 1999) through the execution of conversational actions or strategies such as clarifying questions (Chu-Carroll, 2000; Peng et al., 2018).

**Notation** Consider a goal-oriented conversational environment. Let $\pi_{\theta_i}$ be an LLM's policy parameterized by $\theta$ at timestep $i \geq 0$, with $\pi_{ref}$ being the reference policy model (i.e., $\pi_{ref} \leftarrow \pi_{\theta_0}$). Let $D$ be a dataset consisting of conversations. Let each conversation $c$ in $D$ contain $n$ dialogue turns, through which a user is requesting one or more pieces of information from an agent. The turn state of a conversation (the observed utterances and actions given by each interacting party) at timestep

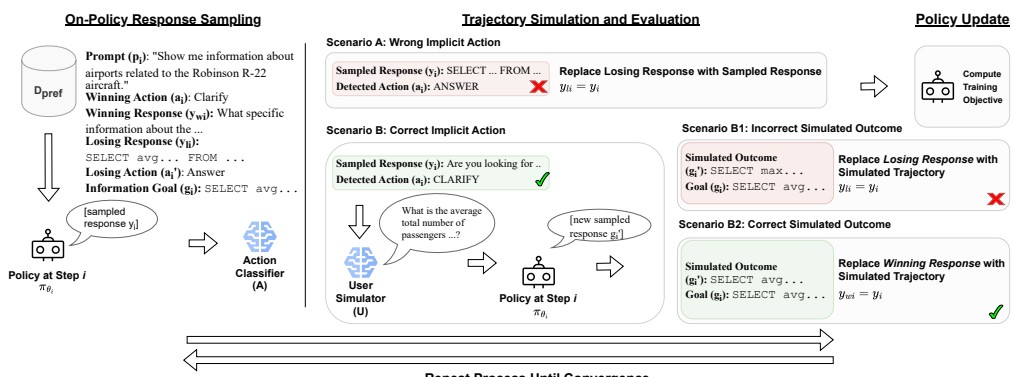

Figure 3: **Overview of the tuning phase of *ACT*.** For each initial contrastive pairing from $D_{pref}$ (constructed as in Sec. 3.2.1), we sample an on-policy response from the model being tuned. After evaluating the sampled response's trajectory, we update the contrastive pairing by either replacing the existing winning or losing response. The model policy is updated using the objective in Eq. 1.

$i$ can be represented by $t_i$. Implicitly, each $t_i$ is part of a trajectory which ends when the user's question expressed at an earlier timestep $j \leq i$ is answered. Any turn $t_i$ can first be broken down into two primary components: $p_i$ and $r_i$, where $p_i$ can be a prompt at $i$, consisting of any task-specific information (e.g. a SQL database schema, tabular data or retrieved passages) combined with any existing dialogue context, and $r_i$ is the ground truth system-side response at $i$. Next, we can let $g_i$ be the goal response which resolves $t_i$'s implicit trajectory, i.e., the answer to the user's original question after any possible clarification turns. In the single-turn trajectory case, $g_i \leftarrow r_i$. Each $r_i$ implicitly expresses an action, $a_i$, where $a_i$ exists in the latent Action Space $S$ of a particular task and $a_i$ can be inferred by some Action Annotation Agent $G$[2]. Thus, we can formally represent turn state $t_i$ using the tuple $(p_i, r_i, g_i, a_i)$. For the datasets considered in our experiments, $S = [\text{CLARIFY, ANSWER}]$ (although the method can be extended to a broader action space). We assume access to a controllable generation model ($M$), Action Classifier ($A$) and model which can be controlled to function as a User Simulator ($U$). As we discuss in Sec. 3.2, $M$ is used for preference data creation whereas $A$ and $U$ are used during tuning and evaluation. We illustrate this notation in Fig. A4.

**User Simulators**    Works such as Deng et al. (2023c); Yu et al. (2023) directly prompt LLMs for goal-oriented tasks conditioned on dialogue context and task objectives. Our implementation of $U$ is inspired by this setup. We first prompt an LLM to summarize the user's information-seeking goal. Then, we form another prompt using this summary along with the current dialogue context to simulate a user response. Prompting with this goal summary allows for more flexibility than directly providing the user simulator with the ground truth information objective. We provide details on our implementation of $U$ in Appendix H.

**Action Classifiers**    In the datasets we considered, the possible actions are to either "clarify" or "directly answer" a question. We directly use few-shot in-context learning as action classifier $A$. We provide details on our in-context examples for $A$ in Appendix G.

## 3.2 *ACT*: ACTION-BASED CONTRASTIVE SELF-TRAINING

One of the north stars in developing intelligent conversational models is the ability to automatically produce responses which take actions which lead to the highest probability of conversational success (Wu et al., 2023; Zhao et al., 2019). We propose *ACT*, an approach that adapts generic LLMs for dialogue generation and models action planning as an implicit subtask. *ACT* is a quasi-online extension of the DPO algorithm which maintains its ease of use of offline method while incorporating the flexible exploration found during online learning. *ACT* relies on a few intuitions. 1) Contrastive preferences are an intuitive medium for demonstrating the *pragmatic differences* between the implicit actions of "winning" and "losing" dialogue responses. 2) Conversational improvements require multi-turn optimization, which are difficult to express using only single-turn contrast pairings. 3) The

---

[2]The gold standard for full label supervision in the context of a fixed dataset is the scenario in which $G$ may be a well-designed human annotation framework such as crowdsourcing. However, at inference time or in settings without label supervision, the implicit action must be inferred by other means such as classification.

gradient of the objective of DPO-like algorithms (see Eq. 2) is weighted based on the log probabilities assigned to the winning and losing responses, and by construction, on-policy response sampling yields high-probability token sequences. *ACT* is summarized in Fig. 3. *ACT* consists of two phases: action-based contrast dataset construction (Alg. 1) and contrastive self-training (Alg. 2).

### 3.2.1 CONSTRUCTION OF PREFERENCE DATA

The preference dataset primarily consists of contrastive *winning-losing* action pairs, as shown in Alg. 1. That is, for each conversation turn $t_i$ in a dataset $D$, we can construct $D_{pref}$ consisting of augmented $t'_i$ tuples. We add rejected action $a'_i$ which is sampled from $S \setminus a_i$, winning response $y_{w_i} \leftarrow r_i$, and $y_{l_i}$ which is a losing response sampled using $M$. Given that each $a'_i$ is predefined when constructing $D_{pref}$, we use a high capacity LLM (Chen et al., 2023b) rather than tuning a smaller one or asking crowdworkers for losing response construction (more details in Appendix F).

---

**Algorithm 1** Building Contrastive Action Pairs

**input** Dataset $D$, Conditional generation model $M$, Action Space $S$, Action Annotation Agent $G$
1: Initialize empty dataset $D_{pref}$.
2: **for** conversation turn $t_i \in D$ **do**
3:     Let $a_i = G(p_i, r_i)$    ▷ Infer Contextual Action
4:     Let $a'_i = S \setminus a_i$   ▷ Determine Rejected Action
5:     Let $y_{wi} = r_i$.
6:     Sample $y_{li} \sim P_M(\cdot|p_i, a'_i)$.
7:     Let $t'_i = (p_i, r_i, g_i, a_i, a'_i, y_{wi}, y_{li})$.
8:     Add $t'_i$ to $D_{pref}$
**output** $D_{pref}$

---

**Action optimization for unlabeled conversations "in-the-wild"** Obtaining gold-standard ambiguity annotations may not always be possible. In such settings, one can obtain pseudo-label supervision using a classifier as the Action Annotation Agent $G$ rather than human annotation. We discuss details and analyze performance in Sec. 5.4. Depending on the data, it may be appropriate to introduce an initial preprocessing step which involves inferring user satisfaction similarly to Shi et al. (2024).

### 3.2.2 SELF-TRAINING USING ON-POLICY CONVERSATION TRAJECTORY SIMULATION

As in DPO training, we continuously sample batches from $D_{pref}$. Although each conversation turn $t_i$ in each batch $j$ has a default winning ($y_{wi}$) and losing ($y_{li}$) response, we also sample an on-policy response $y_i$ from $\pi_{\theta_j}$. We use $A$ to determine whether the implicit action of $y_i$ [3] matches the inferred action $a_i$ of the ground truth response. If the implicit action of $y_i$ is incorrect, we set $y_{li} = y_i$. If it does match $a_i$, then we simulate the outcome $g'_i$ of the trajectory resulting from $y_i$ using $U$ [4] and $\pi_{\theta_j}$. If the trajectory outcome $g'_i$ fails to meet task-specific heuristics (e.g., low semantic similarity or an incorrect execution), we set $y_{li}$ to the entire simulated trajectory resulting from $y_i$ (e.g., "Are you looking for..." + "What is the average total number..." + "SELECT max ..." in Figure 3). Otherwise, we set $y_{wi}$ to the simulated trajectory (e.g. "Are you looking for..." + "What is the average total number..." + "SELECT avg ..." in Figure 3).

### 3.2.3 CONTRASTIVE RL TUNING FOR ALIGNMENT

After constructing the up-to-date winning $y_{wi}$ and losing $y_{li}$ pairing at turn $i$ through simulation (Sec. 3.2.2), we update the policy model ($\pi_\theta$) using the DPO training objective (Rafailov et al., 2024), which is as follows (we ignore the $i$ iterator for simplicity):

$$\mathcal{L}_{\text{DPO}}(\pi_\theta; \pi_{ref}) = -\mathbb{E}_{(p, y_w, y_l) \sim \mathcal{D}} \left[ \log \sigma \left( \beta \log \frac{\pi_\theta(y_w \mid p)}{\pi_{ref}(y_w \mid p)} - \beta \log \frac{\pi_\theta(y_l \mid p)}{\pi_{ref}(y_l \mid p)} \right) \right], \quad (1)$$

where $p$ is a prompt consisting of a concatenation between task info and conversation history = $\{x_1, y_1, ..., x_{i-1}, y_{i-1}, x_i\}$ with each $x_i$ and $y_i$ representing observed user-side and system-side utterances at turn $i$; $y_w$ and $y_l$ are the designated "winning" and "losing" responses or trajectories as set in Sec. 3.2.2; $\pi_{ref}$ is the initial reference policy model; and $\beta$ is a hyperparameter that regularizes the ratio between $\pi_\theta$ and $\pi_{ref}$. The gradient of this objective is given as follows:

$$\nabla_\theta \mathcal{L}_{\text{DPO}}(\pi_\theta; \pi_{ref}) =$$
$$- \beta \mathbb{E}_{(p, y_w, y_l) \sim \mathcal{D}} \left[ \sigma(\hat{R}_\theta(p, y_l) - \hat{R}_\theta(p, y_w)) \left[ \nabla_\theta \log \pi(y_w \mid p) - \nabla_\theta \log \pi(y_l \mid p) \right] \right], \quad (2)$$

---

[3]Classifying $y_i$'s action optimizes the following: $\text{argmax}_{a_k \in S} P_A(a_k|p_i, y_i)$
[4]The next user turn (denoted $u_{i+1}$) is sampled according to $u_{i+1} \sim P_U(\cdot|p_i, y_i)$

---

**Algorithm 2** *ACT*: Action-Based Contrastive Self-Training

---
**input** Initial Policy Model $\pi_{\theta_0}$, Action Contrast Dataset $D_{pref}$, Number of Batches $B$, Action Classifier $A$, User Simulator $U$, Task Heuristic $H$, Heuristic Tolerance $\epsilon$
1:   **for** conversation turn $t_i$ in batch $b_j$ sampled from $D_{pref}$ where $0 \leq j \leq B$ **do**
2:      Sample $y_i \sim P_{\theta_j}(\cdot|p_i)$                         ▷ Sample a response from the current model policy
3:      **if** Action $A(y_i) \neq$ Action $a_i$ **then**
4:         Set $y_{li} = y_i$                         ▷ Implicit pragmatic action does not match ground truth
5:      **else**
6:         Initialize $Trajectory$
7:         Add $y_i$ to $Trajectory$
8:         **while** $A(y_i) \neq ANSWER$ **do**
9:            Clarification Answer $= P_U(p; y_i)$                         ▷ Simulate User Clarification
10:           Add Clarification Answer to $Trajectory$
11:           $y'_{i+1} = P_{\pi_\theta}(P; y_i)$                         ▷ Simulate next policy response
12:           Add $y'_{i+1}$ to $Trajectory$
13:        **if** $H(Trajectory$ outcome, Ground Truth Outcome $g_i) > \epsilon$ **then**
14:           Let $y_{wi} = Trajectory$                         ▷ Reward acceptable trajectory outcome
15:        **else**
16:           Let $y_{li} = Trajectory$                         ▷ Penalize bad trajectory outcome
17:     $\theta \leftarrow Update(\theta)$ until convergence (eq 2)
**output** $\pi_{\theta_B}$

---

where $R(p, y) = \beta \log \frac{\pi(y|p)}{\pi_{ref}(y|p)}$ for a given policy model $\pi$ and reference model $\pi_{ref}$, as proven under the assumptions in Rafailov et al. (2024). The intuition behind the objective is that the gradient of the loss function would increase the likelihood of winning responses $y_w \in Y_w$ and would decrease the likelihood of losing responses $y_l \in Y_l$, with each example being weighed by the magnitude of how incorrectly the implicitly defined reward model ranks the paired responses.

## 4 EXPERIMENTAL SETUP

*ACT* is a sample-efficient approach to adapt an LLM to a conversational action policy. We are primarily concerned with learning optimized implicit selection for agent-side clarification question asking, and we thus evaluate *ACT* as a tuning approach for three complex conversational information-seeking tasks. As a base model for our tuning experiments, we use Zephyr $\beta$, a version of Mistral 7B (Jiang et al., 2023) which has been instruction tuned on UltraChat and aligned to human preferences on UltraFeedback (Cui et al., 2023; Ding et al., 2023; Tunstall et al., 2023).

Table 1: **Experimental results on PACIFIC's public evaluation set**. *ACT* achieves the strongest performance compared to all tuning approaches across every condition in every metric. Tuning-based adaptation strategies are not given any in-context examples at inference time, whereas ICL baselines include 10 in-context conversation examples using the same strategy.

| Adaption Setting | | | Action-level | | Content-level | |
|---|---|---|---|---|---|---|
| Base Model | Approach | Conversations | Macro F1 ↑ | Turn F1 ↑ | Traj. F1 ↑ | Post-Clarify F1 ↑ |
| Gemini Pro | Standard ICL | 10 | 81.4 | 59.7 | 58.7 | **49.7** |
| Claude Sonnet | Standard ICL | 10 | 71.9 | 43.7 | 42.0 | 28.5 |
| Gemini Pro | SFT | 50 | 71.2 | 51.8 | 45.7 | 9.9 |
| Gemini Pro | SFT | 100 | 75.2 | 64.3 | 54.6 | 8.5 |
| Gemini Pro | SFT | 250 | 88.0 | 67.4 | 59.3 | 10.2 |
| Zephyr 7B-$\beta$ | SFT | 50 | 69.0 | 57.8 | 61.3 | 43.5 |
| Zephyr 7B-$\beta$ | IRPO | 50 | 67.7 | 59.1 | 56.7 | 34.4 |
| Zephyr 7B-$\beta$ | *ACT* (ours) | 50 | **82.2** | **62.8** | **61.9** | **57.2** |
| Zephyr 7B-$\beta$ | SFT | 100 | 82.3 | 58.6 | 60.3 | 49.9 |
| Zephyr 7B-$\beta$ | IRPO | 100 | 84.5 | 60.4 | 55.2 | 38.2 |
| Zephyr 7B-$\beta$ | *ACT* (ours) | 100 | **86.0** | **65.0** | **62.0** | **57.4** |
| Zephyr 7B-$\beta$ | SFT | 250 | 86.9 | 65.1 | 63.3 | 56.7 |
| Zephyr 7B-$\beta$ | IRPO | 250 | 85.4 | 64.9 | 58.4 | 40.3 |
| Zephyr 7B-$\beta$ | *ACT* (ours) | 250 | **89.6** | **68.1** | **65.7** | **62.0** |

Table 2: **Abg-CoQA test set evaluation results.** Although Claude Sonnet achieves the highest Action-level performance when prompted with in-distribution in-context conversation examples, it does not result in improved multi-turn goal completion. *ACT* combines on-policy sampling with multi-turn simulation to achieve the best multi-turn goal completion ability on all data settings.

| Adaptation Setting | | | Action-level | Content-level | |
|---|---|---|---|---|---|
| Base Model | Approach | Conversations | Macro F1 ↑ | Turn Similarity ↑ | Traj. Similarity ↑ |
| Gemini Pro | Standard ICL | 10 | 55.5 | **67.0** | **72.2** |
| Claude Sonnet | Standard ICL | 10 | **66.0** | 50.1 | 54.3 |
| Zephyr 7B-$\beta$ | SFT | 50 | 44.6 | 53.3 | 64.2 |
| Zephyr 7B-$\beta$ | *ACT* (ours) | 50 | **52.3** | **66.2** | **68.8** |
| Zephyr 7B-$\beta$ | SFT | 100 | **52.6** | 63.1 | 69.4 |
| Zephyr 7B-$\beta$ | *ACT* (ours) | 100 | 51.1 | **69.5** | **71.4** |
| Zephyr 7B-$\beta$ | SFT | 250 | **53.5** | 64.0 | 66.2 |
| Zephyr 7B-$\beta$ | *ACT* (ours) | 250 | 53.3 | **72.5** | **75.1** |

## 4.1 DATASETS

We investigate three mixed-initiative conversation tasks in which a user interacts with an assistant to retrieve some information. In our setup of each task, a user asks a query which may or may not be underspecified. The assistant is tasked with providing a response which may either be a clarifying question or an attempt to directly answer the user's query. For each task, we synthesize the initial rejected responses by prompting Gemini Ultra as the conditional generation model, $M$. *ACT* is evaluated on a diverse set of datasets spanning various domains: tabular conversational QA, conversational QA for machine reading comprehension, and conversational text-to-SQL generation.

### 4.1.1 PACIFIC: CONVERSATIONAL QA FOR TABULAR DATA

**PACIFIC** is a task for proactive conversational question answering grounded on a mixture of tabular and textual financial data (Deng et al., 2022). This may involve generating the correct words from a given span, from multiple spans, or providing a correct arithmetic expression. The official evaluation for PACIFIC uses a numeracy-focused token overlap metric called DROP F1.

### 4.1.2 ABG-COQA: CONVERSATIONAL QA FOR MACHINE READING COMPREHENSION

**Abg-CoQA** is a conversational question answering dataset for disambiguation in machine reading comprehension (Guo et al., 2021). As there are no arithmetic expressions, we use embedding-based semantic distance with SentenceBERT (Reimers & Gurevych, 2019) as an evaluation metric, which has been used to more flexibly measure question-answering performance (Risch et al., 2021).

### 4.1.3 AMBIGSQL: AMBIGUOUS CONVERSATIONAL TEXT-TO-SQL GENERATION

**AmbigSQL** is our new task for SQL-grounded conversational disambiguation. We systematically perturbed unambiguous queries from Spider, a popular single-turn text-to-SQL benchmark (Yu et al., 2018), resulting in paired training examples which can be easily incorporated into contrastive RL tuning. Each trajectory is evaluated by whether the final proposed SQL query matches the ground truth query's execution result.

Our motivation in constructing AmbigSQL stems from the idea that disambiguation can lead to improved task performance. We prompt an LLM to introduce three types of ambiguous information requests. Those in which the requested information is ambiguous (e.g., "Show details about singers ordered by age from the oldest to the youngest"), those in which the requested population is ambiguous (e.g., "Which ones who live in the state of Indiana?"; see Table A12), and finally, those in which the requested presentation of results is ambiguous (e.g. "Show name, country, age for all singers ordered by age"; see Table A13). We found that constructing a SQL query for an underspecified request with and without clarifications can result in a performance gap of up to 45.8% (see Table A14), demonstrating the necessity of clarifying questions. We provide additional details in Appendix C.

## 4.2 EVALUATION SETUP

We conduct evaluations of *ACT*'s ability to reason about ambiguity in conversation to better accomplish conversational goals along two dimensions.

Table 3: **AmbigSQL test set evaluation**. Zephyr tuned with *ACT* is able to achieve the strongest task performance within each data setting. There are especially large performance improvements in post-clarification SQL execution match when data resources are more scarce.

| | Adaptation Setting | | Action-level | | Content-level | |
|---|---|---|---|---|---|---|
| Base Model | Approach | Conversations | Accuracy ↑ | Macro F1 ↑ | Execution Match ↑ | PC Execution Match ↑ |
| Gemini Pro | Standard ICL | 10 | 72.1 | 70.9 | 63.5 | 75.2 |
| Claude Sonnet | Standard ICL | 10 | 68.5 | 63.8 | 66.5 | 72.4 |
| Zephyr 7B-$\beta$ | SFT | 50 | 77.4 | 77.4 | 21.9 | 13.9 |
| Zephyr 7B-$\beta$ | IRPO | 50 | **91.0** | **91.0** | 27.8 | 30.8 |
| Zephyr 7B-$\beta$ | *ACT* (ours) | 50 | 80.8 | 80.7 | **43.6** | 38.1 |
| Zephyr 7B-$\beta$ | SFT | 100 | 97.2 | 97.2 | 43.3 | 34.3 |
| Zephyr 7B-$\beta$ | IRPO | 100 | 96.2 | 96.1 | 45.0 | 37.0 |
| Zephyr 7B-$\beta$ | *ACT* (ours) | 100 | **99.2** | **99.3** | **48.0** | **49.6** |
| Zephyr 7B-$\beta$ | SFT | 250 | 99.8 | 99.7 | 51.0 | 50.7 |
| Zephyr 7B-$\beta$ | IRPO | 250 | 97.0 | 97.1 | 49.7 | 45.6 |
| Zephyr 7B-$\beta$ | *ACT* (ours) | 250 | **99.9** | **99.8** | **52.3** | **53.0** |
| Zephyr 7B-$\beta$ | SFT | 14,000 (All) | 99.8 | 99.8 | 63.1 | 60.4 |

**Agent task performance:** We evaluate whether *ACT* improves multi-turn task completion capabilities. PACIFIC and Abg-CoQA are initially proposed only with static single-turn evaluations. We mirror this by conducting a turn-level evaluation where we compare the model's response to the ground truth utterance given in response to the user's query, using the task-specific heuristics given in Sec. 4.1. Since we are specifically concerned with improving LLMs' multi-turn capabilities, we additionally propose a multi-turn evaluation scheme for the trajectory outcomes in all three tasks considered. While the sampled response from an LLM is a clarifying question, we simulate a user response and re-sample another response from the evaluated LLM until it attempts to answer the original query. We evaluate this outcome against the user's ground truth information-seeking goal. We use $A$ and $U$ for simulation as described in Sec. 3.2.2 for *ACT*, and use the heuristics defined in Sec. 4.1. An example is illustrated in Fig A5. In PACIFIC and AmbigSQL, we also compute task performance on the simulated responses in which the model has previously asked any clarifying questions, in order to get a more fine-grained measure of the model's ability to reason about its own clarification questions. Details of each evaluation metric for each task are provided in Appendix D.

**Implicit ambiguity recognition**: To help further understand an agent's multi-turn task completion ability, we consider "dialogue act accuracy" (Chen et al., 2023b). Assuming access to ground-truth ambiguity labels, given a contextually-ambiguous user request, a model should generate a clarifying question, otherwise, it should attempt to provide the requested information. We primarily consider Macro F1 since PACIFIC and Abg-CoQA have highly imbalanced classes.

## 4.3 BASELINES

**Prompting baselines** We compare our tuning approaches with smaller models against various prompt-based approaches for multiple frontier LLMs: Gemini 1.5 Pro, Gemini 1.5 Flash, Claude 3.5 Sonnet, and Claude 3.0 Haiku[5]. The results for Gemini Flash and Claude Haiku are included in Appendix B due to space constraints. We use 10 conversations as in-context examples, with three different prompting frameworks: i.) "Standard" which uses the same instruction formatting used for tuning; ii.) chain-of-thought reasoning (Wei et al., 2022); and iii.) "Proactive MIPrompt", the prompting baseline in Deng et al. (2023c), which is a combination of the mixed-initiative prompting approach used in Chen et al. (2023b) and Proactive Prompting (Deng et al., 2023b). We provide a detailed description of each style with examples in Appendix E.

**Tuning baselines** We compare *ACT* with supervised fine-tuning (SFT) as well as other off-policy and on-policy approaches to DPO-based alignment. For SFT, we use the ground truth responses for each dataset's training split. As for DPO-based alignment, an on-policy variant called Iterative Reasoning Preference Optimization (IRPO) was recently proposed and has gained traction for improving model performance in reasoning tasks such as arithmetic. We have thus evaluated IRPO on our two quantitative reasoning tasks, PACIFIC and AmbigSQL. A popular off-policy approach is to sample responses from two high capacity models, with $Y_w$ coming from whichever model is

---

[5] We access each LLM through Vertex AI: https://cloud.google.com/vertex-ai/docs/

Table 4: **Examining ACT on PACIFIC with unlabeled conversational data**. We assume no access to action labels and instead use 0-shot Gemini Pro as the source of action label supervision.

| | Task Adaptation Environment | | | Action-level | Content-level | | |
|---|---|---|---|---|---|---|---|
| Base Model | Framework | Action Supervision | Tuning Ex. | Macro F1 ↑ | Turn F1 ↑ | Traj. F1 ↑ | Post-Clarify F1 ↑ |
| Zephyr 7B-$\beta$ | SFT | NA | 50 | 69.0 | 57.8 | 61.3 | 43.5 |
| Zephyr 7B-$\beta$ | *ACT* | Crowdsourced | 50 | 82.2 | 62.8 | 61.9 | 57.2 |
| Zephyr 7B-$\beta$ | *ACT* | Pseudo-labeled | 50 | 80.1 | 62.4 | 61.1 | 54.7 |
| Zephyr 7B-$\beta$ | SFT | NA | 100 | 82.3 | 58.6 | 60.3 | 49.9 |
| Zephyr 7B-$\beta$ | *ACT* | Crowdsourced | 100 | 86.0 | 65.0 | 62.0 | 57.4 |
| Zephyr 7B-$\beta$ | *ACT* | Pseudo-labeled | 100 | 84.8 | 63.5 | 61.5 | 56.1 |
| Zephyr 7B-$\beta$ | SFT | NA | 250 | 86.9 | 65.1 | 63.3 | 56.7 |
| Zephyr 7B-$\beta$ | *ACT* | Crowdsourced | 250 | 89.6 | 68.1 | 65.7 | 62.0 |
| Zephyr 7B-$\beta$ | *ACT* | Pseudo-labeled | 250 | 89.0 | 68.1 | 64.9 | 61.0 |

of higher capacity (henceforth DPO-Dist; see Mitra et al. (2023); Mukherjee et al. (2023); Xu et al. (2024a)). We present DPO-Dist results in Appendix B.

# 5 EXPERIMENTAL RESULTS

To emulate real-world scenarios with limited data, we evaluate *ACT* as a tuning approach in different scenarios with limited conversation samples across a set of diverse tasks.

## 5.1 CONVERSATIONAL QA WITH TABULAR GROUNDING

In Table 1, we see that across all three data-efficient settings considered, *ACT* achieves the strongest performance across all metrics compared to both SFT and IRPO, which has the advantage of additional test-time computation (Snell et al., 2024; Pang et al., 2024). In particular, *ACT* achieves up to a 19.1% relative improvement over SFT when measuring the tuned model's ability to implicitly recognize ambiguity (from 69.0 to 82.2 Macro F1) given only 50 conversations as tuning data. We also observe that *ACT* has greatly improved data efficiency compared to adapter-based SFT with Gemini Pro, with a relative improvement of as high as 35.7% in multi-turn task performance (from 45.6 to 61.9 in terms of trajectory-level DROP F1). Additionally, tuning with *ACT* in these limited data settings grants the model the ability to match or outperform frontier LLMs used with in-context learning despite having zero in-context examples during inference. Overall, we find that on-policy learning and multi-turn trajectory simulation are crucial for improved multi-turn goal completion.

## 5.2 CONVERSATIONAL QA FOR MACHINE READING COMPREHENSION

Our results for Abg-CoQA are presented in Table 2. In all three data settings, we observe that *ACT* achieved the strongest performance in terms of task-specific metrics (notably, in terms of trajectory-level embedding similarity). However, in the setting with 100 and 250 conversations, Zephyr tuned with SFT slightly outperforms *ACT* in terms of implicit action recognition, although action-level performance primarily helps to contextualize clarification reasoning ability. We discuss this point further in Appendix A. Our approach leads to the strongest turn-level and trajectory-level task performance in all conditions, indicating improved multi-turn reasoning.

## 5.3 CONVERSATIONAL TEXT-TO-SQL GENERATION

We find that although the prompting baselines do not achieve as high Action Accuracy, the benchmarked frontier LLMs can achieve relatively strong downstream performance in terms of execution match. In contrast, tuning Zephyr with both SFT and *ACT* results in rather high Action Accuracy but lower text-to-SQL performance compared to the frontier LLMs. We observe that holistically, *ACT* achieves the largest relative performance improvements in multi-turn task performance compared to other tuning approaches, although the downstream SQL generation ability of larger models is much greater than that of smaller models. This is primarily due to the SQL generation benefiting greatly from scale (Sun et al., 2023b). It is possible that multi-turn performance on larger models can be improved further if *ACT* is applied, as it is even able to yield larger performance improvements than baseline approaches for quantitative reasoning such as IRPO.

## 5.4 *ACT* IN-THE-WILD: LEARNING WITHOUT DIALOGUE ACTION SUPERVISION

Although we have ambiguity labels in the tasks considered here and use them for supervision in Tables 1–3, we also demonstrate that it is possible to perform action-based tuning in the absence

Table 5: **Ablation study of various conditions** using PACIFIC's 50 conversation setting.

| | Macro F1 ↑ | Turn F1 ↑ | Traj. F1 ↑ | Post-Clarify F1 ↑ |
|---|---|---|---|---|
| **Action Importance** | | | | |
| *ACT* w/ Random Actions | 63.2 | 55.3 | 58.7 | 32.8 |
| **Ablation of *ACT* subcomponents** | | | | |
| *ACT* w/o on-policy sampling | 74.8 | 61.5 | 59.1 | 40.5 |
| *ACT* w/ sampling but w/o simulation | 81.4 | 60.8 | 60.2 | 50.1 |
| *ACT* (full) | 82.2 | 62.8 | 61.9 | 57.2 |
| ***ACT* with unaligned foundation models** | | | | |
| Gemma 2B SFT | 57.7 | 38.0 | 40.5 | 17.0 |
| Gemma 2B ACT | **62.7** | **42.6** | **44.0** | **24.8** |
| Mistral 7B SFT | 57.7 | 53.8 | 51.4 | 27.7 |
| Mistral 7B ACT | **75.7** | **58.1** | **57.6** | **31.9** |

of action-label supervision. We use a pre-existing LLM, Gemini 1.5 Pro, as a zero-shot action annotator to re-label the ground truth Assistant-side turns on the PACIFIC corpus. We find that there is astonishingly high agreement (98.5%) with the ground truth action labels. Our results in terms of both Action-level and Content-level metrics reflect that there is nearly no empirical difference in performance. This highlights the potential of *ACT* being highly effective for adaptation to "in-the-wild" settings with smalls amount of unlabeled conversational data.

### 5.5 ABLATION STUDIES

**Are action-based preferences necessary?** One of the key factors of *ACT* is that the contrastive pairs highlight differences between conversational actions. In Table 5 ("*ACT* w/ Random Actions"), we additionally examine the importance of action selection by randomly sampling *both the winning and losing action* when constructing the preference pair, and observe this underperforms normal *ACT*.

**Do we need on-policy sampling?** In Table 5 ("*ACT* w/o on-policy sampling"), we examine the importance of on-policy sampling by evaluating normal off-policy DPO on the dataset as constructed in Sec. 3.2.1. While we do observe some improvements over SFT (e.g., from 69.0 to 74.8 Macro F1), the overall improvements are much larger when using on-policy sampling as with full *ACT*. This may be due to the fact that the off-policy negative responses are not guaranteed to lie in the language manifold of the policy model, and distribution shift may be too difficult to overcome with off-policy learning (Guo et al., 2024).

**Is trajectory simulation necessary?** *ACT* is better-aligned with multi-turn conversations due to its on-policy trajectory simulation. Without multi-turn simulation, our approach can be viewed similarly to on-policy DPO variants like Pang et al. (2024), but with a conversation-specific reward signal which accounts for conversation actions and task heuristics. In Table 5 ("*ACT* w/ sampling w/o simulation"), we find that this trajectory-level simulation is critical to improving multi-turn performance, especially the policy model's ability to reason about its own clarification questions.

**Is *ACT* model agnostic?** The base model in our main experiments, Zephyr, is obtained by aligning Mistral. In Table 5 ("*ACT* with unaligned foundation models") we observe a performance gap of 6.5 Action F1 and 4.3 Trajectory F1 after *ACT* tuning for the two models. However, our results demonstrate *ACT* can improve performance regardless of pre-existing alignment with human feedback, although it can help as an improved model initialization. Overall, we find that improving base model performance with *ACT* is model agnostic.

## 6 CONCLUSION

We propose *ACT*, a model agnostic quasi-online contrastive tuning approach for sample-efficient conversational task adaptation, along with a workflow for evaluation of conversational agents. We demonstrate encouraging evidence that *ACT* is highly effective for task adaptation in the limited data regime, even when there are no action labels available. Future work may also consider combining *ACT* with existing sophisticated tuning approaches for complex tasks like text-to-SQL generation, as well as generalization to large-scale data and multi-task environments.

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

Materialism is a form of philosophical monism which holds that matter is the fundamental substance in nature, and that all phenomena, including mental phenomena and consciousness, are identical with material interactions.

Materialism is closely related to physicalism, the view that all that exists is ultimately physical. Philosophical physicalism has evolved from materialism with the discoveries of the physical sciences to incorporate more sophisticated notions of physicality than mere ordinary matter, such as: spacetime, physical energies and forces, dark matter, and so on. Thus the term "physicalism" is preferred over "materialism" by some, while others use the terms as if they are synonymous. Materialism belongs to the class of monist ontology. As such, it is different from ontological theories based on dualism or pluralism. For singular explanations of the phenomenal reality, materialism would be in contrast to idealism, neutral monism, and spiritualism.

Despite the large number of philosophical schools and subtle nuances between many, all philosophies are said to fall into one of two primary categories, which are defined in contrast to each other: Idealism, and materialism.[a] The basic proposition of these two categories pertains to the nature of reality, and the primary distinction between them is the way they answer two fundamental questions: "what does reality consist of?" and "how does it originate?" To idealists, spirit or mind or the objects of mind (ideas) are primary, and matter secondary. To materialists, matter is primary, and mind or spirit or ideas are secondary, the product of matter acting upon matter.

[Conversation]

User: What would be secondary to them?

Assistant: matter is primary, and mind or spirit or ideas are secondar

User: What does the basic proposition of these two categories pretain to?

Assistant: nature of reality

User: What would materialism be in contrast to

Assistant:

---

**Chosen Response ($y_{wi}$):** Do you want to know what materialism would be in contrast to for singular explanations of the phenomenal reality or for the categories that all philosophies are said to fall into?"

**Chosen Action ($a_i$):** CLARIFY

**Rejected Response ($y_{li}$):** For singular explanations of phenomenal reality, materialism contrasts with idealism, neutral monism, and spiritualism.

**Rejected Action ($a_i'$):** Answer

**Information Goal ($g_i$):** In the context of the two primary philosophical categories, materialism is contrasted with idealism.

---

**Gold Trajectory:**

Assistant: Do you want to know what materialism would be in contrast to for singular explanations of the phenomenal reality or for the categories that all philosophies are said to fall into?

User: For the categories that all philosophies are said to fall into.

Assistant: In the context of the two primary philosophical categories, materialism is contrasted with idealism."

Figure A4: **Example of a contrastive pairing constructed for RL tuning with Abg-CoQA** (Guo et al., 2021). The notation used is as described in Section 3.1.

## A  LIMITATIONS, ETHICAL CONSIDERATIONS, AND BROADER IMPACTS

### A.1  DISCUSSION OF LIMITATIONS

We assume that the clarification questions are appropriately timed. However, crowdsourced conversation datasets are often noisy (Chen et al., 2023b), and relying on noisy annotations or token sequences may result in suboptimal learned policies (from the perspective of asking unnecessary clarifying questions, as well as generating disfluent language). Depending on the source of data, it may be necessary to do an additional preprocessing stage in which one infers whether an action is useful or not. Shi et al. (2024) infers user satisfaction given model responses in-the-wild, whereas Yu et al. (2023) ranks dialogue actions using Monte-Carlo Tree Search.

Label noise can also affect the implicit action recognition evaluation, which assumes that an action in a benchmark task is "optimal." In a corpus like PACIFIC with high inter-annotator agreement (0.62), this is a reasonable assumption. However, we observe inconsistency in Abg-CoQA which may be a result of the low inter-annotator agreement (0.26) reported in Guo et al. (2021). Recent work has demonstrated the promise of many-shot in-context learning Agarwal et al. (2024) compared to supervised fine-tuning at the trade-off of inference-time latency. Yet, Table A6 indicates that even with a greatly increased number of in-context conversation examples (e.g. 50, 100, and 250), the

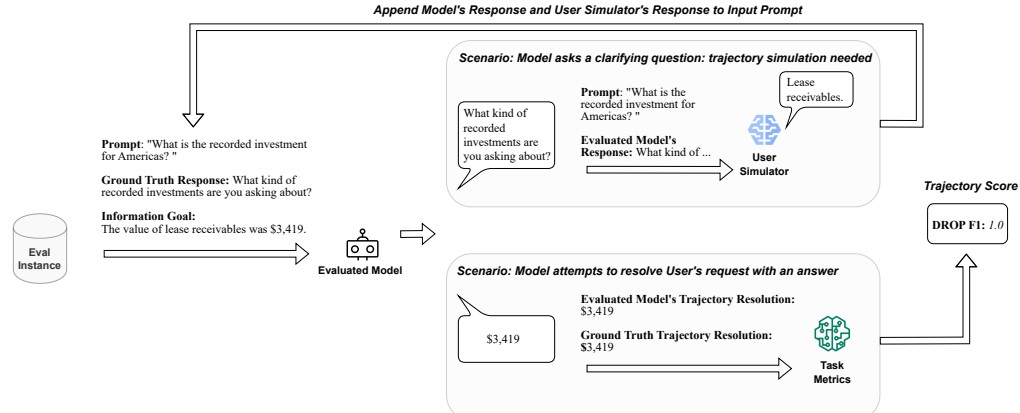

Figure A5: **Trajectory-level content evaluation using the example scenario from Figure 1.** Trajectory-level evaluation seeks to measure the extent to which a candidate LLM can interact with a "User" to reach a target information goal. The "interactive" evaluation of a given instance continues until the candidate LLM attempts to resolve the User's request by providing a direct answer. The candidate trajectory resolution is scored using downstream task metrics. In this example, DROP F1 is used following the task metrics for PACIFIC.

Table A6: **Analysis of the impact of additional data on Abg-CoQA.** Additionaly many-shot examples do not necessarily improve implicit action recognition performance. *ACT* tuning with Zephyr 7B greatly outperforms many-shot Gemini performance.

| Adaptation Setting | | | Action-level | Content-level | |
|---|---|---|---|---|---|
| Base Model | Approach | Conversations | Macro F1 ↑ | Turn Similarity ↑ | Traj. Similarity ↑ |
| Gemini Pro | ICL | 50 | **56.4** | 64.5 | **68.9** |
| Zephyr 7B-$\beta$ | *ACT* (ours) | 50 | 52.3 | **66.2** | 68.8 |
| Gemini Pro | ICL | 100 | **59.2** | 67.0 | **72.0** |
| Zephyr 7B-$\beta$ | *ACT* (ours) | 100 | 51.1 | **69.5** | 71.4 |
| Gemini Pro | ICL | 250 | **58.8** | 66.0 | 71.1 |
| Zephyr 7B-$\beta$ | *ACT* (ours) | 250 | 53.3 | **72.5** | **75.1** |

downstream disambiguation ability does not improve uniformly. We thus posit that in such scenarios, it is more important that for such corpora, multi-turn task completion is a more important measure of a model. We do find that even with 250 in-context examples, tuning a smaller model with *ACT* and the same conversation samples has the potential to outperform frontier models with many-shot examples.

*ACT* also makes use of task-specific heuristics. While this was intentional since success criteria can vary greatly across domains, this may also require more customization and engineering expertise/-effort. Our overall approach to tuning and evaluation also makes heavy use of existing LLMs. We prompt Gemini for purposes such as Action Classification or User Simulation, but such approaches are not perfect and may occasionally result in unwanted behavior. These prompting approaches similarly may require substantial customization efforts. We also realize that not all researchers may have access to commercial LLMs due to researchers for financial or privacy reasons.

Our study also focuses specifically on the limited data regime. We believe that such contexts (e.g., when the target user population is unknown; when conversational data cannot be collected due to privacy concerns; when a conversational system is in its early stages and collecting abundant data for development iteration is not feasible; etc.) would benefit the most from focused adaptation designed to fundamentally teach conversational skills approaches such as *ACT*. As such, in our paper, this was the focus of all of our experiments. It is not clear how much our findings would generalize in settings in which there is an abundance of training data whose distribution closely matches the target

distribution. Intuitively, if much more in-distribution data is made available, even the performance of unaligned objectives like SFT would start to catch up to the performance of focused approaches.

**Is *ACT* online learning?**   Levine et al. (2020) defines offline reinforcement learning as using a fixed dataset of experiences, whereas online reinforcement learning relies on interacting with an environment in real-time. Additionally, Guo et al. (2024) defines on-policy sampling in contrastive RL tuning as settings where both the winning and losing responses are sampled from the policy model. In our case, during tuning, we sample a single response from the policy model.

As such, we define *ACT* as a *quasi-online* contrastive RL tuning approach. *ACT* does rely on action-based preference dataset, as is common in fully-offline reinforcement learning. However, *ACT* continuously samples responses form the policy model in order to update the contrast pairing with good or bad trajectories. Overall, it has both dynamic and static components, so we refer to it as quasi-online. *ACT* also specifically is different from fully online DPO where both the winning and losing responses are sampled (i.e. in Guo et al. (2024)) because our focus is on conversational actions. There is no guarantee that sampling a response from the policy model twice will result in differing actions, unless you change the prompt. However, in that case, you would no longer be computing the DPO objective.

By nature of the domains considered, the extent to which *ACT* allows for online exploration is also limited. As previously mentioned, our experiments are constrained by the fact that there is an objective right/wrong target answer. For instance, if the target answer is an arithmetic expression as is common in PACIFIC, there are a fairly limited number of unique trajectories (when inspected in terms of the number of tokens) that will arrive at that particular expression. In such cases, the trajectory sampled from the policy model during *ACT* tuning may not be any different from the offline trajectory found in the training data.

## A.2   ETHICAL CONSIDERATIONS

**Usage and Safety**   We do not condone the use of our work for any unethical, unlawful, and/or harmful applications. In our work, we do not release any new model artifacts or web-scraped data, but we do not specifically introduce any model guardrails as a part of *ACT*. However, our implementation of *ACT* relies on other LLMs such as Gemini to produce an initial preference dataset, and to perform user simulation. Gemini is released with safety guardrails in place, but these inference-time guardrails may not be available when using open-source LLMs instead. We advise that any deployments of models tuned with *ACT* should consider adding safety guardrails at inference-time.

**Hallucinations**   One commonly documented concern with using LLMs is their tendency to hallucinate answers in Assistant QA contexts. A solution is to provide an LLM with information from a retriever (i.e., retrieval-augmented generation). Two of the datasets we use, PACIFIC and Abg-CoQA, mirror this setting by performing grounded QA using a mixture of long-context textual passages and tabular data. It follows that *ACT* could be further studied in combination with approaches for improved retrieval-augmented generation.

Our evaluation criteria in this paper are also rather restrictive towards hallucinations. In PACIFIC, we use a token-level metric (DROP F1); in Abg-CoQA, we evaluate a candidate response's semantic similarity with a ground truth answer; in AmbigSQL we use execution match, which is a fully objective metric. As such, it is difficult to perform well on PACIFIC and Abg-CoQA if a model consistently hallucinates answers, and in AmbigSQL, a "hallucinated" response would not consist of the appropriate SQL code.

## A.3   BROADER IMPACTS

There is an abundance of modern conversational assistants. *ACT* seeks to improve multi-turn conversational experiences, and thus, it can be used to improve many applications used by potentially millions of users around the world. However, the popularity of conversational assistants also creates an increased risk of misuse. Some people may develop conversational for unethical applications such as misinformation generation, or gray areas such as optimizing dialogue generation for content which is not suitable for the workplace. As discussed above, we do not condone the use of *ACT* for any

Table A7: **Experimental results on PACIFIC's public evaluation set with additional results using Gemini Flash and Claude Haiku**. *ACT* achieves the strongest performance compared to all tuning approaches across every condition in every metric. Tuning-based adaptation strategies are not given any in-context examples at inference time, whereas inference-time adaptation strategies are prompted with 10 in-context conversation examples using the same strategy.

| Adaption Setting | | | Action-level | Content-level | | |
|---|---|---|---|---|---|---|
| Base Model | Approach | Conversations | Macro F1 ↑ | Turn F1 ↑ | Traj. F1 ↑ | Post-Clarify F1 ↑ |
| Gemini Pro | Standard Prompt | 10 | 81.4 | 59.7 | 58.7 | **49.7** |
| Gemini Pro | Chain-of-Thought | 10 | **86.3** | **66.3** | 17.1 | 19.2 |
| Gemini Pro | Proactive MIPrompt | 10 | 78.9 | 63.4 | 61.1 | 18.9 |
| Gemini Flash | Standard Prompt | 10 | 67.4 | 58.8 | 58.7 | 17.9 |
| Gemini Flash | Chain-of-Thought | 10 | 77.1 | 62.0 | 16.9 | 20.0 |
| Gemini Flash | Proactive MIPrompt | 10 | 76.8 | 64.0 | **62.0** | 24.4 |
| Claude Sonnet | Standard Prompt | 10 | 71.9 | 43.7 | 42.0 | 28.5 |
| Claude Sonnet | Chain-of-Thought | 10 | 80.0 | 37.2 | 13.0 | 6.8 |
| Claude Sonnet | Proactive MIPrompt | 10 | 74.9 | 47.2 | 45.9 | 7.6 |
| Claude Haiku | Standard Prompt | 10 | 46.9 | 26.4 | 26.2 | — |
| Claude Haiku | Chain-of-Thought | 10 | 48.6 | 23.7 | 12.0 | 2.9 |
| Claude Haiku | Proactive MIPrompt | 10 | 48.3 | 18.6 | 18.2 | 7.3 |
| Gemini Pro | SFT | 50 | 71.2 | 51.8 | 45.7 | 9.9 |
| Gemini Pro | SFT | 100 | 75.2 | 64.3 | 54.6 | 8.5 |
| Gemini Pro | SFT | 250 | 88.0 | 67.4 | 59.3 | 10.2 |
| Zephyr 7B-$\beta$ | SFT | 50 | 69.0 | 57.8 | 61.3 | 43.5 |
| Zephyr 7B-$\beta$ | DPO-Dist (Pro v. Flash) | 50 | 75.5 | 61.7 | 55.7 | 30.8 |
| Zephyr 7B-$\beta$ | DPO-Dist (Sonnet v. Haiku) | 50 | 74.8 | 62.0 | 56.3 | 31.9 |
| Zephyr 7B-$\beta$ | IRPO | 50 | 67.7 | 59.1 | 56.7 | 34.4 |
| Zephyr 7B-$\beta$ | *ACT* (ours) | 50 | **82.2** | **62.8** | **61.9** | **57.2** |
| Zephyr 7B-$\beta$ | SFT | 100 | 82.3 | 58.6 | 60.3 | 49.9 |
| Zephyr 7B-$\beta$ | DPO-Dist (Pro v. Flash) | 100 | 68.8 | 53.3 | 53.3 | 31.7 |
| Zephyr 7B-$\beta$ | DPO-Dist (Sonnet v. Haiku) | 100 | 83.0 | 59.0 | 53.7 | 29.3 |
| Zephyr 7B-$\beta$ | IRPO | 100 | 84.5 | 60.4 | 55.2 | 38.2 |
| Zephyr 7B-$\beta$ | *ACT* (ours) | 100 | **86.0** | **65.0** | **62.0** | **57.4** |
| Zephyr 7B-$\beta$ | SFT | 250 | 86.9 | 65.1 | 63.3 | 56.7 |
| Zephyr 7B-$\beta$ | DPO-Dist (Pro v. Flash) | 250 | 65.6 | 53.6 | 54.1 | 30.9 |
| Zephyr 7B-$\beta$ | DPO-Dist (Sonnet v. Haiku) | 250 | 82.8 | 43.3 | 38.6 | 19.6 |
| Zephyr 7B-$\beta$ | IRPO | 250 | 85.4 | 64.9 | 58.4 | 40.3 |
| Zephyr 7B-$\beta$ | *ACT* (ours) | 250 | **89.6** | **68.1** | **65.7** | **62.0** |

unethical, harmful, or unlawful applications, and we suggest the usage of safety guardrails for any deployments.

# B  ADDITIONAL EXPERIMENTAL RESULTS

As described in Section 4.3, we include additional prompting experiments using Gemini 1.5 Flash and Claude 3.0 Haiku using Standard Prompting, Chain-of-Thought Prompting, and Proactive Mixed-Initiative Prompting. We also include additional comparisons against an off-policy DPO baseline, DPO-Dist, which involves distilling from two LLMs of differing sizes. The full results for PACIFIC are displayed in Table A7, the full results for Abg-CoQA are in Table A8, and the full results for AmbigSQL are in Table A9.

Table A8: **Abg-CoQA test set evaluation results with additional results using Gemini Flash and Claude Haiku.** *ACT* outperforms SFT across all evaluations in all three data settings. However, Gemini Ultra achieves the strongest downstream task performance when prompted with in-distribution in-context conversation examples.

| | Adaptation Setting | | Action-level | Content-level | |
|---|---|---|---|---|---|
| Base Model | Approach | Conversations | Macro F1 ↑ | Turn Similarity ↑ | Traj. Similarity ↑ |
| Gemini Pro | Standard Prompt | 10 | 55.5 | **67.0** | **72.2** |
| Gemini Pro | Chain-of-Thought | 10 | 61.2 | 63.4 | 39.1 |
| Gemini Pro | Proactive MIPrompt | 10 | 55.5 | 63.3 | 33.3 |
| Gemini Flash | Standard Prompt | 10 | 52.6 | 62.5 | 67.4 |
| Gemini Flash | Chain-of-Thought | 10 | 61.2 | 56.5 | 36.6 |
| Gemini Flash | Proactive MIPrompt | 10 | 58.1 | 61.7 | 36.1 |
| Claude Sonnet | Standard Prompt | 10 | **66.0** | 50.1 | 54.3 |
| Claude Sonnet | Chain-of-Thought | 10 | 63.7 | 46.2 | 36.8 |
| Claude Sonnet | Proactive MIPrompt | 10 | 57.2 | 60.8 | 32.9 |
| Claude Haiku | Standard Prompt | 10 | 49.3 | 40.9 | 41.7 |
| Claude Haiku | Chain-of-Thought | 10 | 46.2 | 30.7 | 28.0 |
| Claude Haiku | Proactive MIPrompt | 10 | 45.2 | 34.5 | 31.4 |
| Zephyr 7B-$\beta$ | SFT | 50 | 44.6 | 53.3 | 64.2 |
| Zephyr 7B-$\beta$ | DPO-Dist (Pro v. Flash) | 50 | 46.9 | 57.2 | 61.2 |
| Zephyr 7B-$\beta$ | DPO-Dist (Sonnet v. Haiku) | 50 | 44.7 | 57.9 | 61.5 |
| Zephyr 7B-$\beta$ | *ACT* (ours) | 50 | **52.3** | **66.2** | **68.8** |
| Zephyr 7B-$\beta$ | SFT | 100 | **52.6** | 63.1 | 69.4 |
| Zephyr 7B-$\beta$ | DPO-Dist (Pro v. Flash) | 100 | 47.8 | 61.9 | 67.1 |
| Zephyr 7B-$\beta$ | DPO-Dist (Sonnet v. Haiku) | 100 | 44.8 | 62.0 | 66.4 |
| Zephyr 7B-$\beta$ | *ACT* (ours) | 100 | 51.1 | **69.5** | **71.4** |
| Zephyr 7B-$\beta$ | SFT | 250 | **53.5** | 64.0 | 66.2 |
| Zephyr 7B-$\beta$ | DPO-Dist (Pro v. Flash) | 250 | 46.0 | 61.9 | 66.3 |
| Zephyr 7B-$\beta$ | DPO-Dist (Sonnet v. Haiku) | 250 | 46.3 | 62.6 | 67.0 |
| Zephyr 7B-$\beta$ | *ACT* (ours) | 250 | **53.3** | **72.5** | **75.1** |

Table A9: **AmbigSQL test set evaluation with additional results using Gemini Flash and Claude Haiku**. Zephyr tuned with *ACT* is able to achieve the strongest task performance within each data setting. There are especially large performance improvements in post-clarification SQL execution match when data resources are more scarce.

| | Adaptation Setting | | Action-level | Content-level | |
|---|---|---|---|---|---|
| Base Model | Approach | Conversations | Accuracy ↑ | Execution Match ↑ | PC Execution Match ↑ |
| Gemini Pro | Standard Prompt | 10 | 72.1 | 63.5 | 75.2 |
| Gemini Flash | Standard Prompt | 10 | 75.6 | 64.2 | 66.2 |
| Claude Sonnet | Standard Prompt | 10 | 68.5 | 66.5 | 72.4 |
| Claude Haiku | Standard Prompt | 10 | 73.8 | 57.3 | 65.3 |
| Zephyr 7B-$\beta$ | SFT | 50 | 77.4 | 21.9 | 13.9 |
| Zephyr 7B-$\beta$ | DPO-Dist (Pro v. Flash) | 50 | 77.7 | 42.6 | 31.5 |
| Zephyr 7B-$\beta$ | DPO-Dist (Sonnet v. Haiku) | 50 | 78.0 | 40.9 | **41.2** |
| Zephyr 7B-$\beta$ | IRPO | 50 | 91.0 | 27.8 | 30.8 |
| Zephyr 7B-$\beta$ | *ACT* (ours) | 50 | **80.8** | **43.6** | 38.1 |
| Zephyr 7B-$\beta$ | SFT | 100 | 97.2 | 43.3 | 34.3 |
| Zephyr 7B-$\beta$ | DPO-Dist (Pro v. Flash) | 100 | 98.7 | 45.1 | 45.3 |
| Zephyr 7B-$\beta$ | DPO-Dist (Sonnet v. Haiku) | 100 | **99.8** | 47.8 | 44.8 |
| Zephyr 7B-$\beta$ | IRPO | 100 | 96.2 | 45.0 | 37.0 |
| Zephyr 7B-$\beta$ | *ACT* (ours) | 100 | 99.2 | **48.0** | **49.6** |
| Zephyr 7B-$\beta$ | SFT | 250 | 99.8 | 51.0 | 50.7 |
| Zephyr 7B-$\beta$ | DPO-Dist (Pro v. Flash) | 250 | 97.3 | 49.7 | 44.2 |
| Zephyr 7B-$\beta$ | DPO-Dist (Sonnet v. Haiku) | 250 | 99.7 | 50.7 | 50.3 |
| Zephyr 7B-$\beta$ | IRPO | 250 | 97.0 | 49.7 | 45.6 |
| Zephyr 7B-$\beta$ | *ACT* (ours) | 250 | **99.9** | **52.3** | **53.0** |
| Zephyr 7B-$\beta$ | SFT | 14,000 (All) | 99.8 | 63.1 | 60.4 |

## C  *AmbigSQL*: MODELING *Ambig*UITY IN CONVERSATIONAL TEXT-TO-*SQL*

Table A10: **Overview of AmbigSQL**, an ambiguous Text-to-SQL dataset synthesized from Spider.

|  | Train | Dev | Test |
|---|---|---|---|
| Num. Unambiguous Requests | 7,000 | 1,034 | 1,034 |
| Num. Ambiguous Requests | 7,000 | 1,034 | 1,034 |
| Num. Unique Schemas | 1,056 | 145 | 145 |
| Types of Ambiguity | 3 | 3 | 3 |

There is growing interest in using LLM-based agents for coding tasks (Liu et al., 2024). Particularly, due to the complexity of such tasks, multi-turn interactions in which an agent is able to clarify assumptions and user intents should intuitively help with goal completion (Nijkamp et al., 2023). Despite this, there are few existing resources for multi-turn code generation tasks. One example is CoSQL, a conversational text-to-SQL task which also includes linguistic ambiguities (Yu et al., 2019), but the proposed task does not include agent-style interaction in which a model must learn to ask clarifying questions. Upon our inspection of the dataset, there are also various inconsistencies related to the "system-side" clarification questions given in the dataset's conversational contexts, which we highlight in Table A11. As a result, we propose AmbigSQL, our own synthetically constructed resource for ambiguous conversational text-to-SQL.

Table A11: **Conversations in CoSQL with noisy "clarification questions"** (highlighted in red). Example 1) is a remnant of crowdsourcing in which the system-side party makes mention of the task guideline. Example 2) demonstrates a system-side clarification question being asked prior to the user making any information requests. Example 3) The system-side clarification question makes reference to some prior database search result, but the execution feedback is not made accessible to the system during inference.

| No. | Interacting Party | Utterance |
|---|---|---|
|  | User | Can you list all the singer ids that aren't present in the song table? |
|  | Assistant | `SELECT Name FROM singer WHERE Singer_ID NOT IN ...` |
|  | User | Thanks! |
| 1 | Assistant | You should ask at least 3 questions |
| 2 | Assistant | Did you want the full name of makers and the number? |
| 3 | Assistant | Do you mean the address of the customer with first name Luis? |

### C.1  AMBIGSQL CONSTRUCTION

We start from Spider, a popular single-turn text-to-SQL dataset and benchmark. An overview of AmbigSQL is given in Table A10. Due to the nature of the single-turn requests in Spider, each instance can be viewed as a conversation consisting of a single state $t_1$ (see notation defined in Section 3). In $t_1$, $p_1$ contains any instructions, the database schema, and the user's request. $r_1$ is the correct SQL query which yields the information requested by the user. $g_1 = r_1$ because the trajectory ends on the same turn due to the system yielding the correct query. $a_1 = ANSWER$ because the only action possible is to provide a SQL query.

Our desired result in constructing AmbigSQL is to have a corpus which can be used to demonstrate to an LLM the linguistic differences between ambiguous and clear user requests. That is, given a fixed database schema and a target SQL query, we want a pair of requests such that one requires asking a clarification question and one does not. This would also result in a balanced dataset in which half of the requests require asking clarification questions, and half do not.

We focus on three fundamental types of ambiguous information requests. Those in which the requested information is ambiguous (e.g., "Show details about singers ordered by age from the oldest to the youngest"), those in which the requested population is ambiguous (e.g., "Which ones who live

Table A12: **In-context example given as part of a prompt for creating information requests in which the target population is ambiguous.** The format of the black text represents how a ground-truth request would be used to form the prompt for a target example. The blue text represents the content that would be synthesized from an LLM. We omit the database schema from the paper.

[Database Schema Omitted]
The target SQL query is the following:

```
SELECT professional_id ,  last_name ,  cell_number FROM Professionals
WHERE state  =  'Indiana' UNION SELECT T1.professional_id ,  T1.last_name ,
T1.cell_number FROM Professionals AS T1 JOIN Treatments AS T2 ON
T1.professional_id  =  T2.professional_id
GROUP BY T1.professional_id HAVING count(*)  >  2
```

Here is a clear request that would correspond to this SQL query:
"Which professionals live in the state of Indiana or have done treatment on more than 2 treatments? List his or her id, last name and cell phone."
Here is the same request converted into an ambiguous format by underspecifying the target columns:
"Which ones live in the state of Indiana or have done treatment on more than 2 treatments?"
Here is an appropriate clarifying question to recover the clear request from the ambiguous request:
"Are you asking about the Professionals?"

---

in the state of Indiana?"; see Table A12), and finally, those in which the requested presentation of results is ambiguous (e.g. "Show name, country, age for all singers ordered by age"; see Table A13).

We iterate through each of the examples in Spider, and use an LLM (Gemini Ultra 1.0) to synthesize a perturbed version of each unambiguous query, along with an appropriate clarifying question. For queries which require some manipulation of results presentation, we prompt an LLM to perturb the query such that the requested presentation style becomes ambiguous. Otherwise, we randomly select a perturbation strategy: either masking the requested information (Table A13) or masking the requested population (Table A12). For each of these three strategies, we use five in-context examples demonstrating the appropriate masking process. The exact in-context examples are given in the attached code. Each of these ambiguous queries are thus associated with conversations containing "ground truth" states at two timesteps, $t_1$ and $t_2$. $p_1$ contains an ambiguous user request, $r_1$ is the synthesized clarification question, and accordingly, $a_1 = CLARIFY$. $p_2$ contains the disambiguated user request, $r_2$ is the correct ground truth SQL query, and accordingly, $a_2 = ANSWER$.

The code to create AmbigSQL will be released publicly. Each instance includes the database metadata included in Spider, but for all experiments used in this paper, the database schema is linearized into the format used for prompting in Sun et al. (2023a;b).

---

Table A13: **In-context example given as part of a prompt for creating information requests in which the target columns are ambiguous.** The format of the black text represents how a ground-truth request would be used to form the prompt for a target example. The blue text represents the content that would be synthesized from an LLM. We omit the database schema from the paper.

[Database Schema Omitted]
The target SQL query is the following:

```
SELECT professional_id ,  last_name ,  cell_number FROM Professionals
WHERE state  =  'Indiana' UNION SELECT T1.professional_id ,  T1.last_name ,
T1.cell_number FROM Professionals AS T1 JOIN Treatments AS T2 ON
T1.professional_id  =  T2.professional_id
GROUP BY T1.professional_id HAVING count(*)  >  2
```

Here is a clear request that would correspond to this SQL query:
"Which professionals live in the state of Indiana or have done treatment on more than 2 treatments? List his or her id, last name and cell phone."
Here is the same request converted into an ambiguous format by underspecifying the target columns:
"Which professionals live in the state of Indiana or have done treatment on more than 2 treatments?"
Here is an appropriate clarifying question to recover the clear request from the ambiguous request:
''Which information of the professionals do you want to know?"

Table A14: **Examination of ambiguity in AmbigSQL.** Competitive high capacity LLMs struggle with producing "correct" SQL queries given only ambiguous user requests. Including disambiguation turns in the prompts greatly improves execution match.

| Model | Ambiguous Request Execution Match | Execution Match with Clarification Turns |
|---|---|---|
| Gemini Pro | 28.5 | 68.7 |
| Gemini Ultra | 31.2 | 77.0 |

## C.2 EXAMINING CLARIFICATION NEED IN AMBIGSQL

We primarily are concerned with examining the extent to which clarification questions are necessary in providing the requested SQL queries. Inspired by the notion of "recovery" presented in Toles et al. (2023), we examine the performance in constructing each of the unique SQL queries with and without the gold clarification turns. Concretely, we first evaluated the Execution Match performance achieved by prompting LLMs with only the ambiguous versions of each user request, with each instruction including the instruction that the LLM must construct a SQL query. Then, we prompted LLMs to construct the same SQL queries, but given the disambiguation turns as conversation history (i.e., with context consisting of the original ambiguous request, the clarification question, and then non-ambiguous request).

We conducted this analysis on the test set using two competitive LLMs, Gemini Pro and Gemini Ultra, with the Execution Match tool from Spider and CoSQL (Yu et al., 2018; 2019). Our results are shown in Table A14. Given only an ambiguous request, both Gemini Pro and Gemini Ultra struggle to consistently construct the correct SQL query. However, given disambiguation turns, Execution Match improves dramatically, approximating the performance on the validation set given in Sun et al. (2023a;b).

## C.3 EXAMPLES

Table A15 and Table A16 each contain a pair of examples from AmbigSQL's test set. Each example contains the prompt which is provided to an LLM, the immediate ground truth response to the user request provided as part of the prompt, and the resulting ground truth trajectory (for examples which include clarifying questions). The examples in each table are paired — the top example is directly taken from Spider and converted into a conversational format, and the bottom example is the result of introducing ambiguity into the first example.

Table A15: **Examples from the AmbigSQL test set using the "concert_singer" database.** Black text: context provided to an LLM as a prompt. Blue text: ground-truth single-turn response. Purple text: target trajectory which should be achieved through simulation.

[Instruction]
You are an Assistant having a conversation with a User who would like to retrieve some information from a SQL database with the schema provided below. If you are confident in the User's intent, you should provide the appropriate SQL query. Otherwise, you should ask the User a clarifying question.
[Database Schema Omitted]
[Conversation]
User: How many singers do we have?
Assistant: SELECT count(*) FROM singer

[Instruction]
You are an Assistant having a conversation with a User who would like to retrieve some information from a SQL database with the schema provided below. If you are confident in the User's intent, you should provide the appropriate SQL query. Otherwise, you should ask the User a clarifying question.
[Database Schema Omitted]
[Conversation]
User: Tell me about the singers.
Assistant: What specifically would you like to know about the singers? For example, would you like to know how many there are, or see a list of their names?
User: How many singers do we have?
Assistant: SELECT count(*) FROM singer

Table A16: **Examples from the AmbigSQL test set using the "flight_2" database.** Black text: context provided to an LLM as a prompt. Blue text: ground-truth single-turn response. Purple text: target trajectory which should be achieved through simulation.

[Instruction]
You are an Assistant having a conversation with a User who would like to retrieve some information from a SQL database with the schema provided below. If you are confident in the User's intent, you should provide the appropriate SQL query. Otherwise, you should ask the User a clarifying question.
[Database Schema Omitted]
[Conversation]
User: Return the number of airports.
Assistant: SELECT count(*) FROM AIRPORTS

[Instruction]
You are an Assistant having a conversation with a User who would like to retrieve some information from a SQL database with the schema provided below. If you are confident in the User's intent, you should provide the appropriate SQL query. Otherwise, you should ask the User a clarifying question.
[Database Schema Omitted]
[Conversation]
User: How many are there?
Assistant: Could you please specify which table you are referring to when you ask 'How many are there?'
User: Return the number of airports.
Assistant: SELECT count(*) FROM AIRPORTS

## D  ADDITIONAL DETAILS ON EVALUATION

We define the following action-level evaluation metrics:

- Accuracy: Percentage of correct implicit actions
- Macro F1: Unweighted Average of F1 for each action

We define the following content-level evaluation metrics:

- Turn-level DROP F1: Average immediate response DROP F1 Deng et al. (2022); Dua et al. (2019)
- Trajectory-level DROP F1: Average trajectory-outcome DROP F1 Deng et al. (2022); Dua et al. (2019)
- Post-Clarification DROP F1: Average DROP F1 Deng et al. (2022); Dua et al. (2019) of responses which follow agent clarification turns
- Turn-level Similarity: Immediate response embedding similarity
- Trajectory-level Similarity: Trajectory outcome embedding similarity
- Trajectory-level Execution Match: Percentage of trajectory outcomes with correct execution results
- Post-Clarification Execution Match: Percentage of trajectory outcomes with correct execution results out of those that which contain clarification turns

**PACIFIC**  As described in Section 4, PACIFIC is a conversational question-answering dataset in which the final answers may involve generating the correct words from a given span, from multiple spans, or providing a correct arithmetic expression. As such, the authors propose using DROP F1 as the official evaluation metric. The way DROP F1 is used in the original paper Deng et al. (2022) is analogous to our aforementioned "Turn-level DROP F1." However, as this does not fully represent a model's conversational reasoning abilities, we additionally evaluate LLMs in the PACIFIC environment using Macro F1, Trajectory-level DROP F1, and Post-Clarification DROP F1. Concretely, the evaluation for some LLM $\pi$ is as follows. Assume we have some example with prompt $p$, winning action $a$, ground truth response $r$, and trajectory-level information goal $g$. We sample a candidate response from the LLM: $y \sim P_\theta(\cdot|p)$. We then simulate the trajectory resulting from each response $y$ according to Lines 6-12 in Algorithm 2 and obtain trajectory outcome $g'$. The aforementioned action-level metrics are computed using the implicit actions of each $y$ with each ground truth implicit action $a$. Turn-level DROP F1 is computed between all sampled responses $y$ and all ground truth responses $r$, and Trajectory-level DROP F1 is computed over all simulated trajectory outcomes $g'$ and all ground truth information goals $g$. Post-Clarification F1 is defined as Trajectory-level F1 for only the subset of trajectories which include clarification turns.

**Abg-CoQA**  As previously mentioned in Section 4, Abg-CoQA is a conversational question-answering dataset for machine reading comprehension. Thus, we use embedding similarity Risch et al. (2021) as it allows for producing more coherent and diverse responses which may be scored lowly by criteria such as token-level F1 score. In the original paper, language models are only evaluated in terms of QA performance Guo et al. (2021), rather than their ability to disambiguate requests. Thus, for our evaluation, we remove all clarification turns from the prompt and require the LLM to produce clarifying questions on its own. However, unlike the other tasks considered in this paper, each ambiguous request is paired with all of the possible trajectories (i.e., the reasons why the request is considered ambiguous). We thus perform an evaluation for every ground truth trajectory, so that it is impossible for an LLM to achieve a high cumulative trajectory-level score simply by getting lucky at guessing the user's intent.

Concretely, the evaluation for some LLM $\pi$ is as follows. Assume we have some example with prompt $p$, winning action $a$, ground truth response $r$, and the set of trajectory-level information goals $G$. For every individual trajectory-level goal $g \in G$, we sample a candidate response from the LLM: $y \sim P_\theta(\cdot|p)$ then simulate the trajectory resulting from each response $y$ according to Lines 6-12 in Algorithm 2 and obtain trajectory outcome $g'$. As with PACIFIC, we compute Macro F1 using the implicit actions of each $y$ with each ground truth implicit action $a$. We compute Turn-level similarity for *each unique* $p$ between sampled responses $y$ and ground truth responses $r$. We compute

Table A17: **"Standard" in-context example given as part of a prompt to an LLM for response generation in PACIFIC.** The purple text is the overall system instruction. The blue text is what an LLM would have to produce at inference time.

---

You are an Assistant answering questions from a User. You should either attempt to answer the question or ask a clarifying question if there is any ambiguity.
[Table and Passage Omitted]
User: How much does the company expect to make contributions to the defined plans during the next fiscal year?
Assistant: ['What kind of defined plans are you asking about?']
User: The defined benefit plans and the defined contribution plan respectively.
Assistant: ['$5.1 million', '$0.6 million']
User: What is the value of the expected defined contribution plan as a percentage of the defined benefit plan during the next fiscal year?
Assistant: 0.6/5.1 = 11.76
User: How much would change if there is a 1% increase and a 1% decrease respectively in the discount rate?
Assistant: ['What kind of change are you asking about?']
User: The pension.
Assistant: ['$(39,145)', '$49,361']
User: What is the difference in that amount if the discount rate had a 1% increase and a 1% decrease?
Assistant: 49,361 - (39,145) = 88506
User: How much would the pension change if there is a 1% increase and a 1% decrease respectively?
Assistant: ['Which rate of increase and decrease are you talking about?']
User: The future salary growth.
Assistant: ['$7,572', '$(6,919)']
User: What is the difference between them?
Assistant: 7,572 - (6,919) = 14491

---

Trajectory-level similarity over all simulated trajectory outcomes $g'$ and all ground truth information goals $g$.

**AmbigSQL** is a ambiguous text-to-SQL generation task. By design, the labels are balanced, so we only use Accuracy for action-level metrics. The primary objective is to produce SQL queries which result in correct executions, so we evaluate LLMs in terms of Trajectory-level and Post-Clarification Execution Match. For each instance in the test set, we sample a response from an LLM and compute the trajectory as specified in Lines 6-12 in Algorithm 2. Trajectory-level Execution Match includes SQL queries which are constructed with or without clarification turns, whereas Post-Clarification Execution Match only looks at SQL queries which are constructed following clarification turns.

# E    DESCRIPTION OF IN-CONTEXT LEARNING BASELINES

We use several in-context learning baselines with frontier LLMs in Section 4. For each condition, we randomly sample 10 conversation examples from each task's 250-instance data pool, and apply one of the following prompting frameworks.

**Standard Prompting**    We simply provide in-context examples that are structurally identical to the inputs used for model tuning. Our format is similar to the formats used in Chen et al. (2023b); Deng et al. (2023b) and we provide an example in Table A17.

**Chain-of-Thought Prompting**    We integrate the popular reasoning framework, chain-of-thought prompting (Wei et al., 2022) into our aforementioned "Standard" conversational prompt format. Effectively, we ask an LLM to do end-to-end dialogue generation by having it first produce a reasoning chain which states whether the current context is ambiguous or not. We provide an example in Table A18.

**Proactive Mixed-Initiative Prompting**    Following the baselines given in Deng et al. (2023c), we apply the Proactive Prompting framework Deng et al. (2023b) mixed with the Mixed-Initiative Prompting style Chen et al. (2023b). Ultimately, the LLM conditions on the possible set of actions,

Table A18: **"Chain-of-Thought" in-context example given as part of a prompt to an LLM for response generation in PACIFIC.** The purple text is the overall system instruction. The blue text is what an LLM would have to produce at inference time.

You are an Assistant answering questions from a User. You should either attempt to answer the question or ask a clarifying question if there is any ambiguity.
[Table and Passage Omitted]
User: What is the value of the expected defined contribution plan as a percentage of the defined benefit plan during the next fiscal year?
Instruction: If the user's question is ambiguous, ask an appropriate clarifying question. Otherwise, directly answer the user's question using the information from the passage context and the table. Let's think step by step.
Reasoning: The user's question is not ambiguous. Assistant: 0.6/5.1 = 11.76
User: How much would change if there is a 1% increase and a 1% decrease respectively in the discount rate?
Instruction: If the user's question is ambiguous, ask an appropriate clarifying question. Otherwise, directly answer the user's question using the information from the passage context and the table. Let's think step by step.
Reasoning: The user's question was ambiguous. Assistant: ['What kind of change are you asking about?']
User: The pension.
Instruction: If the user's question is ambiguous, ask an appropriate clarifying question. Otherwise, directly answer the user's question using the information from the passage context and the table. Let's think step by step.
Reasoning: The user's question is not ambiguous. Assistant: ['$(39,145)', '$49,361']
User: What is the difference in that amount if the discount rate had a 1% increase and a 1% decrease?
Instruction: If the user's question is ambiguous, ask an appropriate clarifying question. Otherwise, directly answer the user's question using the information from the passage context and the table. Let's think step by step.
Reasoning: The user's question is not ambiguous. Assistant: 49,361 - (39,145) = 88506 User: How much would the pension change if there is a 1% increase and a 1% decrease respectively?
Instruction: If the user's question is ambiguous, ask an appropriate clarifying question. Otherwise, directly answer the user's question using the information from the passage context and the table. Let's think step by step.
Reasoning: The user's question was ambiguous.
Assistant: ['Which rate of increase and decrease are you talking about?']

along with interweaved natural language instructions that describe which actions correspond to existing dialogue turns. We provide an example in Table A19. We use this framework for each of the long-context dialogue corpora (PACIFIC and Abg-CoQA).

Table A19: **"Proactive Mixed-Initiative" in-context example given as part of a prompt to an LLM for response generation in PACIFIC.** The purple text is the overall system instruction. The blue text is what an LLM would have to produce at inference time.

You are an Assistant answering questions from a User. You should either attempt to answer the question or ask a clarifying question if there is any ambiguity.
[Table and Passage Omitted]
User: How much does the company expect to make contributions to the defined plans during the next fiscal year?
The user's last question was ambiguous. The Assistant asks a clarifying question.
Assistant: ['What kind of defined plans are you asking about?']
User: The defined benefit plans and the defined contribution plan respectively.
The user's last question was unambiguous. The Assistant directly answers the question.
Assistant: ['$5.1 million'$, '$0.6$ million']
User: What is the value of the expected defined contribution plan as a percentage of the defined benefit plan during the next fiscal year?
The user's last question was unambiguous. The Assistant directly answers the question.
Assistant: 0.6/5.1 = 11.76
User: How much would change if there is a 1% increase and a 1% decrease respectively in the discount rate?
The user's last question was ambiguous. The Assistant asks a clarifying question.
Assistant: ['What kind of change are you asking about?']
User: The pension.
The user's last question was unambiguous. The Assistant directly answers the question.
Assistant: ['$(39, 145)'$, '$49,361$']
User: What is the difference in that amount if the discount rate had a 1% increase and a 1% decrease?
The user's last question was unambiguous. The Assistant directly answers the question.
Assistant: 49,361 - (39,145) = 88506
User: How much would the pension change if there is a 1% increase and a 1% decrease respectively?
Actions: ["Directly Answer", "Ask a Clarification Question"]
Prompt: Given the task background and the conversation history, please use appropriate actions to generate the response.
Response: The user's last question was ambiguous. The Assistant asks a clarifying question.
Assistant: ['Which rate of increase and decrease are you talking about?']

Table A20: **In-context example given as part of a prompt to an LLM for action classification with PACIFIC.** The blue text represents the action for this example.

---

[Table and Passage Omitted]
User: What was the total number of acquisitions in FY19?
Assistant: 35 acquisitions
User: What are the benefits of active growth strategy?
Assistant: strengthens and scales the National Storage operating platform which drives efficiencies across the business.
User: What was the total NLA?
Assistant: Which region are you asking about?
The last Assistant utterance is a clarifying question.

---

Table A21: **In-context example given as part of a prompt to an LLM for action classification with Abg-CoQA.** The blue text represents the action for this example.

---

[Passage Omitted]
User: And who in the group was anxious?
Assistant: Peppe
User: Was she well-rested?
Assistant: no
The last Assistant utterance is a direct answer.

---

## F  CONDITIONAL GENERATION MODEL DETAILS

As mentioned Section 3.1, we make use of a high capacity LLM as a conditional generation model, $M$. For all experiments considered, we use Gemini Ultra to construct the initial action-based contrastive preference dataset. We follow the format of "mixed-initiative prompting" described in Chen et al. (2023b), rather than performing zero-shot inference, we use ten in-context examples to adapt the LLM to the prompting structure. Concretely, for a given input/output pair to be used as an in-context example, we interweave each System-side utterance with a narrative instruction that states that the next utterance is either a clarifying question or a direct answer. This yields "control" over the pragmatic action of the generated utterance.

## G  ACTION CLASSIFIER DETAILS

Compared to recognizing whether a request is ambiguous or not, action classification (i.e., recognizing whether an existing utterance is a question or answer attempt) is considerably simpler. We directly prompt Gemini Ultra 1.0 with 10 in-context examples to serve as $A$ (as per the notation given in Section 3.1).

Table A20 is an example of an in-context example used to demonstrate a system-side response with the "Clarify" action in PACIFIC ("Assistant: Which region are you asking about?"). All of the conversation history is made available in addition to the relevant table and passage in the prompt. We include all contextual information available as part of the input to the Action Classifier, along with the last Assistant response which is an example of the classification target.

Table A21 is an example of an in-context example used to demonstrate a system-side response with the "Answer" action in Abg-CoQA ("Assistant: no"). As is common in this corpus, a subset of the conversation is made available in addition to the relevant passage in the prompt. We include all contextual information available as part of the input to the Action Classifier, along with the last Assistant response which is an example of the classification target.

Table A22 is an example of an in-context example used to demonstrate a system-side response with the "Answer" action in AmbigSQL. In AmbigSQL, the information provided as input is the database schema, and the user's initial information request. This information is provided as part of the input to the Action Classifier, along with the last Assistant response which is an example of the classification target.

Table A22: **In-context example given as part of a prompt to an LLM for action classification with AmbigSQL.** The blue text represents the action for this example.

---

[Database Schema Omitted]
User: "Which professionals live in the state of Indiana or have done treatment on more than 2 treatments? List his or her id, last name and cell phone."
Here is the same request converted into an ambiguous format by underspecifying the target columns:
Assistant:

```
SELECT professional_id ,  last_name ,  cell_number FROM Professionals
WHERE state  =  'Indiana' UNION SELECT T1.professional_id ,  T1.last_name ,
T1.cell_number FROM Professionals AS T1 JOIN Treatments AS T2 ON
T1.professional_id  =  T2.professional_id
GROUP BY T1.professional_id HAVING count(*)  >  2
```

The last Assistant utterance is a direct answer.

---

## H  USER SIMULATOR DETAILS

In order to iterate on system-side dialogue models, many works investigate the use of a user simulator, $U$ (as per the notation given in Section 3.1) (e.g. Chandramohan et al. (2011); Kreyssig et al. (2018); Shah et al. (2018)). At its core, it requires a response generation model which can robustly represent user-side intents. Traditionally, this involved training a model specifically on user-side responses but due to the aforementioned control capabilities of modern LLMs Chen et al. (2023b), we directly perform few-shot prompting for user simulation, following Deng et al. (2023c); Yu et al. (2023). For a given conversation during training, we again prompt a high-capacity LLM with in-context examples to identify the information-seeking user intents. Then, conditioned on the generated information-seeking intents, we prompt the same LLM with the responses from the policy model to simulate the sub-goal trajectory. This allows $U$ to ground on the real intents of the user while retaining enough flexibility to respond coherently to some system-side response. In our work, we use Gemini Ultra 1.0 with three hand-crafted in-context examples for both intent summarization and user response generation.

Table A23: **In-context example given as part of a prompt to an LLM for intent summarization with PACIFIC.** The purple text represents the system prompt used for this task. The blue text represents the summarized information-seeking intents.

---

The following is a conversation between a User and an Assistant. The User is asking some questions. Summarize what information the User is looking for.
[Table and Passage Omitted]
User: What does Walletron deliver?
Assistant: patented mobile wallet technology.
User: How much did it cost the company to acquire certain technology assets of RevChip, LLC ("RevChip") and TranSend Integrated Technologies Inc. ("TranSend")?
Assistant: $7.0 million
User: What was the pro forma revenue in 2019?
Assistant: $1,382,957
User: What was the change in its amount between 2018 and 2019?
Assistant: 21228
User: How about that in pro forma net income?
Assistant: -6425
[Information] The user wants to know: 1. What technology Walletron delivers, 2. How much it cost Walletron to acquire certain technology assets of RevChip, LLC and TranSend Integrated Technologies Inc., 3. What the pro forma revenue was in 2019, 4. What the change in pro forma revenue was between 2018 and 2019, and 5. What the change in pro forma net income was between 2018 and 2019.

---

**PACIFIC**  We provide one of our three handcrafted in-context examples of summarizing information-seeking intent in Table A23. While the information summary is handcrafted, the conversation and grounding data are taken directly from the PACIFIC. After intent summarization, $U$ is given the existing conversation history along with an instruction that includes the information

seeking intents and asked to provide the next user response. This format is structurally the same as the three simulation-specific in-context examples — an example is provided in Table A24.

Table A24: **In-context example given as part of a prompt to an LLM for user simulation with PACIFIC.** The purple text represents the system prompt used for this task, which contains the information-seeking intents summarized from Table A23. The entire conversation is used for this example to demonstrate grounding between a conversation and intents.

The following is a conversation between a User and an Assistant. The User is asking some questions. The user wants to know: 1. What technology Walletron delivers, 2. How much it cost Walletron to acquire certain technology assets of RevChip, LLC and TranSend Integrated Technologies Inc., 3. What the pro forma revenue was in 2019, 4. What the change in pro forma revenue was between 2018 and 2019, and 5. What the change in pro forma net income was between 2018 and 2019.
[Table and Passage Omitted]
Assistant: $7.0 million
User: What was the pro forma revenue in 2019?
Assistant: $1,382,957
User: What was the change in its amount between 2018 and 2019?
Assistant: 21228
User: How about that in pro forma net income?
Assistant: -6425

**Abg-CoQA** We conduct user simulation for Abg-CoQA similarly to PACIFIC. Table A25 is one of three hand-crafted examples of intent summarization, and Table A26 is one of three hand-crafted examples used for simulating the final user response.

Table A25: **In-context example given as part of a prompt to an LLM for intent summarization with Abg-CoQA.** The purple text represents the system prompt used for this task. The blue text represents the summarized information-seeking intents.

[Passage Omitted]
User: What was his ranking?
Assistant: General
User: Did someone else have horse fighters?
Assistant: yes
User: Who?
Assistant: Do you want to know who had 2,500 horse fighters initially?
User: No, I want to know who had a considerable force of horse fighters west of him.
Assistant: General (Bishop) Polk.
Instruction: Summarize what information the user is looking for.
Summary: The user wants to know: 1. What his ranking was. 2. Whether someone else had horse fighters. 3. Who had a considerable force of horse fighters west of him.

**AmbigSQL** We conduct user simulation slightly differently for AmbigSQL. Rather than having a separate step for knowledge-seeking intent summarization, we directly condition $U$ on the target SQL query. As demonstrated in Table A27, the prompt includes the target SQL query, the original user request, and the clarification question sampled from the Assistant. In the example given in Table A27, the last line is the ground truth disambiguated query, and in practice, this would be the generation target of $U$.

# I QUALITATIVE EXAMPLES

In Table A28 we compare two trajectories produced by Zephyr 7B tuned on 50 conversations from PACIFIC using SFT and *ACT*, respectively, given the same conversation context and tabular information. The SFT model provided an incorrect answer to the user request, matching the "Incorrect Assumption" case in Figure 1. In contrast, the model tuned using *ACT* asked a clarifying question before eventually producing the correct answer.

Table A26: **In-context example given as part of a prompt to an LLM for user simulation with Abg-CoQA.** The purple text represents the system prompt used for this task, which contains the information-seeking intents summarized from Table A25. The entire conversation is used for this example to demonstrate grounding between a conversation and intents.

The following is a snippet from a conversation between an Assistant and a User who is asking some questions about the Story below. The user only wants to know the following information: 1. What his ranking was. 2. Whether someone else had horse fighters. 3. Who had a considerable force of horse fighters west of him.
[Passage Omitted]
User: What was his ranking?
Assistant: General
User: Did someone else have horse fighters?
Assistant: yes
User: Who?
Assistant: Do you want to know who had 2,500 horse fighters initially?
User: No, I want to know who had a considerable force of horse fighters west of him.
Assistant: General (Bishop) Polk.

Table A27: **In-context example given as part of a prompt to an LLM for user simulation with AmbigSQL.** The purple text represents the system prompt used for this task, which contains ground truth SQL query goal. The entire conversation is used for this example to demonstrate SQL query-grounded request disambiguation.

A user is asking an assistant to retrieve some information from a SQL database. The command that the assistant should ultimately return is as follows:

```
SELECT county FROM campuses where campus = 'California State University-Chico'
```

The assistant will ask some questions to clarify the user's intent. The user should respond with a rephrased request that reflects their desired query.
User: what is the county?
Assistant: Are you asking for a list of all of the counties in the database?
User: I'm looking for the county of the campus 'California State University-Chico'

Table A28: **Comparison of trajectories using different tuning approaches**. Left: snippet of tabular data from the PACIFIC evaluation set. Middle: trajectory resulting from the SFT model in response to the user query. Right: trajectory resulting from the *ACT* model. Red text: sampled from $U$.

| Relevant Table Data | SFT | ACT |
|---|---|---|
| Year: 2019 ‖ 2018 | User: What were the total liabilities of IMFT? | User: What were the total liabilities of IMFT? |
| Total current liabilities : 851 ‖ 167 | Assistant: *$909* ✗ | Assistant: Which year are you asking about? |
| Other noncurrent liabilities : 5 ‖ 74 | | User: 2018 |
| Total Liabilities: $909 ‖ $1,305 | | Assistant: *$1,305* ✓ |

Table A29: **Comparison of PACIFIC trajectories using different tuning approaches**. Left: snippet of from a passage from the PACIFIC test set. Middle: trajectory resulting from the SFT model in response to the user query. Right: trajectory resulting from the *ACT* model. Red text: sampled from $U$. Both the SFT and *ACT* model answered the question incorrectly.

| | SFT | ACT |
|---|---|---|
| **Relevant Data** | User: What was the net profit/(loss) after tax? | User: What was the net profit/(loss) after tax? |
| Net profit/(loss) after tax was $(9.8) million | Assistant: $(9,819) × | Assistant: Which year are you asking the amount of? |
| Net profit/(loss) after tax : (9,819) ‖ 6,639 ‖ (248%) | | User: FY19 |
| | | Assistant: $(9,819) × |

Table A30: **Comparison of PACIFIC trajectories using different tuning approaches**. Left: snippet of from a passage from the PACIFIC test set. Middle: trajectory resulting from the SFT model in response to the user query. Right: trajectory resulting from the *ACT* model. Red text: sampled from $U$. Both the SFT and *ACT* model answered the question incorrectly, and neither asked a clarification question.

| | SFT | ACT |
|---|---|---|
| **Relevant Table Data** | User: In which year was the amount of Investments higher? | User: In which year was the amount of Investments higher? |
| Year: 2019 ‖ 2018 | Assistant: [2018] × | Assistant: [2018, 2019] × |
| Investments: 1,216.0 ‖ 1,212.9 | | |

Table A29 includes an erroneous trajectory produced by the *ACT* model in the 50 conversation setting of PACIFIC. In this particular example, both the provided passage and the provided table provided similar information regarding net profit/loss after tax. In the dataset, the ground truth answer selected a span from the passage, whereas the *ACT* model produced a clarification question and then eventually an answer using tabular information. The SFT model directly assumed the intent of the user and similarly produced an answer from the tabular information.

Table A30 includes another pair of erroneous trajectories in the 50 conversation setting of PACIFIC. The requested information was accessible through the provided table data, but both the SFT model and the *ACT* model failed to produce the correct answer. This example was not annotated as requiring a clarifying question, so it simply indicates that there is much room for improvement in terms of downstream task performance.

Table A31 includes a pair of trajectories resulting from SFT and *ACT* in the 50 conversation setting of Abg-CoQA. The requested information is given in the portion of the passage in the left panel. The SFT model asked an off-topic clarification question and failed to bring the qestioning on track. The *ACT* model successfully asked an on-track clarification question and eventually produced the correct answer.

Table A31: **Comparison of Abg-CoQA trajectories using different tuning approaches**. Left: snippet of a relevant passage from the Abg-CoQA test set. Middle: trajectory resulting from the SFT model in response to the user query. Right: trajectory resulting from the *ACT* model. Red text: sampled from $U$. Ground truth outcome: Meghan asked Lizzie,"Are you awake?"

| | SFT | ACT |
|---|---|---|
| **Relevant Passage** | User: What did Meghan ask? | User: What did Meghan ask? |
| She yawned, stretching out like a lazy cat. | Assistant: Do you mean where did Lizzie see her father? | Assistant: Do you mean that morning or the night before? |
| Her sister was also awake. "Lizzie", | User: No | User: The night before. |
| Meghan whispered, "are you awake?" | Assistant: Lizzie had only met her father in person once, according to the story. × | Assistant: Meghan asked Lizzie if she was awake that morning. ✓ |

# J  ADDITIONAL EXPERIMENTAL DETAILS

## J.1  COMPUTE RESOURCES

We conduct all experiments using one Google Compute Engine Virtual Machine with 8x 80GB A100 GPUs. All of our tuning experiments use full-model multi-GPU tuning with Deepspeed. Full *ACT* with 250 conversations (1,923 training examples) from PACIFIC took the longest amount of time and required approximately 16 hours to run for 12 epochs.

Beyond the experiments reported in the paper, there were many preliminary experiments that took place as a part of this overall research project that were ultimately directionally different from our final contribution.

### J.2 TRAINING HYPERPARAMETERS

For all of our SFT experiments with Zephyr, Mistral, and Gemma, we tune the model for up to 8 epochs. We choose the best-performing model with learning rates from $\{1e-4, 2e-5, 1e-5\}$ with the AdamW optimizer.

For our SFT experiments with Gemini Pro, we use the Vertex AI API[6] and tune for up to 4 epochs with an Adapter size of 4.

For all of our RL tuning experiments, we allow the model to train for up to 12 epochs, and select the checkpoint that results in the highest reward margin on the validation set (which is an action-based preference dataset constructed as described in Section 3.2 using each task's original validation set). For all experiments, we use a batch size of 4, and a maximum sequence length of $1,280$.

**Hyperparameters for Equation 2** For experiments with Zephyr 7B on PACIFIC, we achieve our strongest results using $\beta = 0.01$ and a learning rate of $5e-7$. On AmbigSQL, we use $\beta = 0.01$ and a learning rate of $5e-7$. On AmbigSQL, we use $\beta = 0.5$ and a learning rate of $5e-7$.

## K ASSETS USED

All resources used have been cited appropriately in the paper. In this section, we enumerate each of the existing artifacts used in our work along with their license.

**Existing Models**

- Gemma Gemma Team et al. (2024): Gemma Open-Source License. `https://ai.google.dev/gemma/terms`
- Gemini 1.0 Ultra (gemini-1.0-ultra), Gemini 1.5 Pro (gemini-1.5-pro-001), Gemini 1.5 Flash (gemini-1.5-flash-001) (Gemini Team et al., 2023): Accessed through the Google Cloud Vertex AI Platform. `https://cloud.google.com/products/gemini?hl=en`
- Claude 3.5 Sonnet, Claude 3.0 Haiku (Anthropic AI, 2024): Accessed through the Google Cloud Vertex AI Platform. `https://cloud.google.com/products/gemini?hl=en`
- MiniLM-L6-v2 (Reimers & Gurevych, 2019): Apache 2.0. `https://huggingface.co/sentence-transformers/all-MiniLM-L6-v2`
- Mistral 7B-v0.1 (Jiang et al., 2023): Apache 2.0. `https://huggingface.co/mistralai/Mistral-7B-v0.1`
- Zephyr 7B-$\beta$ (with Mistral 7B as a Base Model) (Tunstall et al., 2023): MIT Open-Source License. `https://huggingface.co/HuggingFaceH4/zephyr-7b-beta`

**Existing Datasets**

- Abg-CoQA (Guo et al., 2021): MIT Open-Source License. `https://github.com/MeiqiGuo/AKBC2021-Abg-CoQA`
- PACIFIC (Deng et al., 2022): MIT Open-Source License. `https://github.com/dengyang17/PACIFIC/tree/main`
- Spider (Yu et al., 2018): CC BY-SA 4.0. `https://yale-lily.github.io/spider`

**Existing Algorithms and Software**

---

[6]`https://cloud.google.com/vertex-ai/generative-ai/docs/models/gemini-use-supervised-tuning`

- Direct Preference Optimization (Rafailov et al., 2024): CC BY 4.0.
- Google Cloud Pipeline Components: Apache 2.0. `https://cloud.google.com/vertex-ai/docs/pipelines/components-introduction`
- HuggingFace Transformers (Wolf et al., 2020): Apache 2.0. `https://github.com/huggingface/transformers/tree/main`
- PyTorch (Paszke et al., 2019): PyTorch Open Source License. `https://github.com/pytorch/pytorch/tree/main`
- Vertex AI SDK: Apache 2.0. `https://cloud.google.com/vertex-ai/docs/python-sdk/use-vertex-ai-python-sdk`

