# OpenReview forum: "Learning to Clarify: Multi-turn Conversations with Action-Based Contrastive Self-Training"
_ICLR.cc/2025/Conference — ICLR 2025 Poster_

### Official Review · Reviewer_thTU · 2024-11-01

**Soundness:** 2
**Presentation:** 2
**Contribution:** 3
**Rating:** 5
**Confidence:** 3

**Summary:**

The paper proposes Action-Based Contrastive Self-Training (ACT), to improve large language models' (LLMs) conversational skills, particularly in handling ambiguity and learning optimal dialogue policies. The paper shows results in three conversational tasks, including tabular-grounded question answering, machine reading comprehension, and text-to-SQL generation, highlighting the importance of considering action-based preferences for conversational tasks.

**Strengths:**

Originality
The paper proposes a novel DPO algorithms for handle ambiguity, which to the best of my knowledge is novel in this settings.

Clarity
The paper is overall clear (Algorithm 2 is pretty helpful to understand the overall process).

**Weaknesses:**

Significance
Low overall significance. The experiments focus on a specific sub-domain of dialogue systems (SQL-based), and a particular skill, which larger language model already posses. Moreover, the results focus on a single LLM, and report dubious SFT performance. This diminish the significance of the proposed approach.

**Questions:**

- Why is the SFT results reported in Table 1 and 2 do not improve with more data? Can you try a cross validation split?

**Details Of Ethics Concerns:**

Nan

---

> ### Author Response · Authors · 2024-11-22
> **Response to Reviewer thTU (1/2)**
>
> Thank you so much for taking the time to review our paper, and for acknowledging the novelty of our work as well as the clarity of our paper! We appreciate your helpful feedback which has greatly helped to strengthen our paper, and we have done our best to respond to your questions below.
>
> > **“The experiments focus on…”**
>
> > **”a particular skill which larger language model already posses.”**
>
> R: Thank you for raising this point! While large LLMs have excelled at many difficult tasks, we would like to clarify that unfortunately, even larger LLMs *do not currently possess the ability by default to proactively lead conversations by asking clarification questions*. This type of skill is important to learn in adaptation to downstream tasks, where interactivity may be more desirable (e.g. specialized applications such as building a dialogue system for customer support; coding assistants), and we observe *ACT is more effective for downstream task adaptation than generic strategies such as SFT or in-context learning*.
>
> In our Action F1 results in Table 1 of our paper (we note that there is high inter-annotator agreement on PACIFIC, indicating ambiguity here is mostly objective), and we have now run additional benchmarking experiments with zero-shot dialogue context as input at inference time and longer adapter tuning on Gemini to further evaluate this ability:
>
> | Condition | Action F1 |
> |---------------|---------------|
> | Gemini Pro 0-Shot (new) | 58.1 |
> | Gemini Pro 10-Shot inference | 81.4 |
> | Gemini 50 conversations SFT (new) | 71.2 |
> | Gemini 100 conversations SFT (new) | 75.2 |
> | Gemini 250 conversations SFT | 88.0 |
> | Zephyr 10-Shot inference (new) | 55.4 |
> | Zephyr 250 conversation SFT, 0-shot inference | 86.9 |
> | Zephyr 250 conversation ACT, 0-shot inference | 89.6 |
>
> Prompting Gemini zero-shot with specific instructions for disambiguation *only achieves an Action F1 of 58.1.* Gemini Pro achieves an Action F1 of 81.4 using 10 full in-context conversation examples at inference time. Implicit ambiguity recognition performance can be improved further with Gemini using 250 samples for SFT, *reaching 88.0 F1 without any additional inference-time cost.*
>
> We have additionally run experiments with 10-shot in-context learning on a smaller LLM, Zephyr 7B, finding that the model only achieves 55.4 Action F1. We find that with 250 conversations for tuning, *Zephyr tuned with ACT reaches as high as 89.6 Action F1, outperforming Gemini tuned with SFT.*
>
> Overall, with Zephyr, we consistently observe the performance ranking as ACT > SFT > few-shot ICL. With Gemini, there is not a commercial interface that allows us to run ACT, but we see the potential of SFT is higher than few-shot ICL. *Overall, we find that specialized tuning approaches may have a higher ceiling for effective LLM customization than generic prompting approaches as they are better aligned with downstream tasks. This has the potential to greatly reduce inference time latency/complexity* (e.g., reducing the need for many-shot prompting).
>
> > **”a specific sub-domain of dialogue systems (SQL-based)”**
>
> R: We thank you for your valuable suggestion on making our paper more impactful on various domains! First and foremost, we do agree with you that a method is more useful when it can be applied to various domains. Following this reason, our work has been evaluated on a variety of domains covering finance, news articles, childrens stories, etc., and feature complex reasoning subtasks over both structured tables and natural language documents. Our work is **not** restricted to SQL-specific subdomains. Please refer to Table 1, which demonstrates our experimental results on PACIFIC which *mixes structured tables with natural language*, and Table 2, our Abg-CoQA results which involve reasoning about *natural language documents*.
>
> Secondly, we would also like to mention that text-to-SQL is an important task which has received much attention in recent years both in academia (e.g. [1,2,3,4,5]) and industry (e.g. [6,7,8,9]). Our SQL-based conversational question answering results are only found in Table 3.
>
> We have also run limited experiments on the planning task of another general goal-oriented dialogue corpus [10] to demonstrate that our approach can generalize to interactive settings with *non-conversational actions like tool use*, featuring a non-binary action space. The results below highlight that *our method achieves a higher average reward even on this action optimization task that does not contain explicit clarification labels in the data.*
>
> | Condition | Normalized Reward |
> |---------------|---------------|
> | Zephyr SFT | 0.538 |
> | Zephyr ACT | 0.592 |

---

> > ### Author Response · Authors · 2024-11-22
> > **Response to Reviewer thTU (2/2)**
> >
> > > **”The SFT results reported in Table 1 and 2 do not improve with more data”**
> >
> > The Zephyr SFT results reported in Table 1 strictly improve with additional data.
> >
> > For our results on Abg-CoQA in Table 2, we have re-run baselines with an expanded hyperparameter search space for SFT and updated the results accordingly. There is more consistent improvement on this corpus, and the results are summarized below for convenience:
> >
> > | Condition | Macro F1 | Single-Turn Sim | Multi-Turn Sim |
> > |---------------|---------------|---------------|---------------|
> > | Zephyr 7B 50 sample SFT | 44.6 | 53.3 | 64.2 |
> > | Zephyr 7B 50 sample ACT | 52.3 | 66.2 | 68.8 |
> > | Zephyr 7B 100 sample SFT | 52.6 | 63.1 | 69.4 |
> > | Zephyr 7B 100 sample ACT | 51.1 | 69.5 | 71.4 |
> > | Zephyr 7B 250 sample SFT | 53.5 | 64.0 | 66.2 |
> > | Zephyr 7B 250 sample ACT | 53.3 | 72.5 | 75.1 |
> >
> > However, we would also like to mention there is some label noise on Abg-CoQA specifically due to relatively low inter-annotator agreement (0.26 Cohen’s Kappa). This noise may lead to unstable performance with offline learning (e.g. SFT), which relies solely on this fixed dataset.  In contrast, PACIFIC has 0.62 IAA, and ambiguity is introduced by design in AmbigSQL. We have further verified this label noise on Abg-CoQA through many-shot in-context learning with Gemini, finding that *only increasing the number of conversation samples from Abg-CoQA does not lead to consistent improvement in action recognition*.
> >
> > On AbgCoQA:
> > | Condition | Macro F1 | Single-Turn Sim | Multi-Turn Sim |
> > |---------------|---------------|---------------|---------------|
> > | Gemini Pro 50-Shot | 56.4 | 64.5 | 68.9 |
> > | Gemini Pro 100-Shot | 59.2 | 67.0 | 72.0 |
> > | Gemini Pro 250-Shot | 58.8 | 66.0 | 71.1 |
> > | Zephyr 7B 250 sample ACT, zero-shot inference | 53.3 | 72.5 | 75.1 |
> >
> > Overall, we would like to emphasize that the primary objective of goal-oriented conversation environments is that the agent should *navigate the interaction to provide the requested information*. Action recognition performance is helpful to interpret the goal-completion ability of an agent but can be impacted by label noise on the benchmark task. To this end, the multi-turn content evaluation most directly represents this objective. We find that the exploration encouraged by the quasi-online aspect of ACT may help to overcome the label noise in the original dataset as exhibited by the strong multi-turn performance. Having 250 in-context examples at inference time should provide an advantageous inductive bias for Gemini Pro, but we are able to surpass the single-turn similarity (72.5 versus 66.0) and multi-turn similarity (75.1 versus 71.1) on this task using ACT tuning with Zephyr, a much smaller model. We have added these results and emphasized these discussions in Section 4.2, Appendix A.1, Table A6 of the revised manuscript.
> >
> > > **“Moreover, the results focus on a single LLM”**
> >
> > In Tables 1, 2, and 3, we have experimental results for prompting with both Gemini and Claude as requested. As ACT is a tuning approach, *our primary experiments demonstrate the improvements of ACT compared to tuning baselines on diverse tasks*, which we have clearly demonstrated with Zephyr 7B. In Table 5, we additionally used ACT tuning with other base LLMs: we considered both Gemma 2B and Mistral 7B, finding that *ACT can improve model performance significantly compared to SFT regardless of the model*.
> >
> > We thank you again for your very helpful feedback and hope we have addressed each of your concerns!
> >
> > References:
> >
> > [1] Can LLM Already Serve as A Database Interface? A Big Bench for Large-Scale Database Grounded Text-to-SQLs, NeurIPS 2023 https://arxiv.org/abs/2305.03111
> >
> > [2] L2CEval: Evaluating Language-to-Code Generation Capabilities of Large Language Models, TACL 2024 https://direct.mit.edu/tacl/article/doi/10.1162/tacl_a_00705/124835
> >
> > [3] CHESS: Contextual Harnessing for Efficient SQL Synthesis https://arxiv.org/abs/2405.16755
> >
> > [4] MAC-SQL: A Multi-Agent Collaborative Framework for Text-to-SQL https://arxiv.org/abs/2312.11242
> >
> > [5] Granite Code Models: A Family of Open Foundation Models for Code Intelligence https://arxiv.org/abs/2405.04324
> >
> > [6] https://www.snowflake.com/en/data-cloud/cortex/
> >
> > [7] https://www.databricks.com/product/ai-bi/genie
> >
> > [8] https://repost.aws/articles/ARJszlMEepRti6xoM-0fsBmw/bringing-generative-ai-to-the-data-warehouse-with-amazon-bedrock-and-amazon-redshift
> >
> > [9] https://cloud.google.com/blog/products/data-analytics/introducing-data-qna
> >
> > [10] Decision-oriented dialogue for human-ai collaboration, TACL 2024 https://arxiv.org/abs/2305.20076

---

> > > ### Author Response · Authors · 2024-11-24
> > >
> > > Dear Reviewer thTU,
> > >
> > > We thank you again for your valuable feedback! Your comments have helped us to further investigate the existing limitations in conversational abilities of both small and large LLMs without specialized tuning. You have also helped us to clarify the significance of our work which examines three impactful conversational domains, and led us to demonstrate that our proposed ACT approach is also applicable to domains with broad, non-conversational actions. Your feedback has also helped us to refine our explanation of the purpose of our evaluation metrics in the paper.
> > >
> > > We respectfully ask whether you may have any remaining concerns preventing any appropriate score adjustments, and we will do our best to address them.
> > >
> > > Thank you again for your time!
> > >
> > > Kind regards,
> > >
> > > The Authors

---

> > > > ### Author Response · Authors · 2024-11-26
> > > >
> > > > Dear Reviewer thTU,
> > > >
> > > > We thank you again for your invaluable insights! You have helped us to greatly strengthen our paper through an investigation which has led to highlighting the limitations of existing LLMs, clarifying the significance of our work across diverse domains, demonstrating the generalizability of our proposed ACT approach, and improving the discussion of our evaluation metrics in the paper.
> > > >
> > > > As the discussion period comes to a close, we respectfully ask whether you may have any remaining questions or concerns that would prevent any appropriate score adjustments, and we will do our best to address them in the remaining time.
> > > >
> > > > Thank you again for your time and your helpful feedback!
> > > >
> > > > Kind regards,
> > > >
> > > > The Authors

---

### Official Review · Reviewer_Qb6Q · 2024-11-03

**Soundness:** 4
**Presentation:** 3
**Contribution:** 3
**Rating:** 8
**Confidence:** 3

**Summary:**

The authors present ACT, a framework for fine-tuning LLMs to respond better to ambiguous queries. The approach relies on online preference optimization, using a DPO-like objective, where the user responses and action classifications are provided by some other LLM user simulation and a trained action classifier, and the agent responses are provided by the model under tuning. The authors design a particular, detailed approach to generating conversational examples from which the model learns, and show that it is highly effective and with relatively few examples enables mid-size (~7B parameters) LMs to outperform very large LLMs even with 10 examples in their prompts. Finally, the authors provide a new dataset of ambiguous queries that are resolved to SQL queries, again showing that the ACT framework provides better results by learning to ask clarifying questions.

Updated ratings after reading author responses. In particular the presentation has improved and the paper is easier to understand. Thank you to the authors for their efforts.

**Strengths:**

First and foremost, the authors present extensive comparisons showing that their approach leads to better conversational outcomes, both in terms of recognizing the need to clarify a user prompt, and in resolving the user's information need. The difference is sometimes quite large, with 10-20% improvements over SFT or a modified DPO approach. It is a significant result, showing that off-policy learning techniques can be bested by even this quasi-online learning approach. While not alone in finding this, the authors make an important contribution to showing the limits of off-policy learning and the result is timely as more and more effort is focused on trying to get better performance out of more reasonably sized models. The addition of the new AmbigSQL data set should prove useful as it is more easily evaluated automatically.

**Weaknesses:**

The paper is probably over-stuffed with results, and is generally very wordy making it difficult to read. It is saved by the clear algorithms and figures, but the text should be shortened considerably. This would also allow for more discussion of the AmbigSQL data set, which is a significant contribution, but which is barely described in the text. As an example, I found that the first three sentences of the paragraph at L350 ("Implicit Ambiguity Recognition") all say essentially the same thing. This could be shortened to make room for more informative text. Similarly, I don't know that we need to see every possible result in the main results tables. Focus on the main results/models to tell your story, and omit the rest to make the tables easier to read. I didn't find the ablation experiments useful in this case--you could have summarized the entire section in one short paragraph.

The AmbigSQL data might be too easy to learn, as some models achieve near perfect performance on it.

The paper is novel, but not exceptionally so. One thing you could do after shortening the rest of the paper is to consider the use of user simulators more, and find parallels between earlier work and yours. I think the novelty of the paper lies mostly in the exact implementation of data generation for preference optimization, summarized in Algorithm 1 and Figure 3, and it would be good to focus on that specifically.

**Questions:**

Figure 3 was critical to my understanding of your approach. It would help the text to label figure 3 with the notation used in the text, and to refer to specific parts of the figure in the text to better connect the two. Without the figure I would have been completely lost in section 3.

I found all of the sections on "conversations in the wild" difficult to follow and kind of disconnected from the rest of the text. Can you clarify what specific problem you are trying to address with these tests?

In Algorithm 2 you say (line 14-16) that the _trajectory_ is assigned to the winning or losing labels--but in the text it is just the sampled y_j that is assigned to these depending on the conversation outcome. These should me made consistent with each other.

I would like to see details of the user simulator U and action classifier A in the main text. Making your text less wordy and shortening the tables to just the key results will help make room to discuss these critical pieces more. The other results from the tables, and things like the ablations, can be expounded on more fully in the appendices. That's what appendices are for: extra results and things that don't turn out to be as interesting.

You could also omit most or all of the DPO-Dist results from the main text--it never gets close to the best results. You can mention this in the main text and then detail it more in the appendices.

Line 304: How do you guarantee that the responses from Gemini Ultra model M _should_ be rejected? Won't it often come up with a good response?

Line 313: I admit I haven't focused on this aspect for some time, but surely there's something better than SentenceBERT for evaluation now?

Line 354: why switch metrics between datasets? Why not just stick with F1, even though your AmbigSQL dataset is balanced? It seems like being consistent would help us get a better sense of how challenging (or not) the AmbigSQL dataset is.

Lines 425-427: ACT does better than frontier LLMs when it has 250 examples to learn from. I assume these are generated examples, and therefore you don't need human labeling. It makes me wonder (1) how much does this depend on the models you use being updated with human labeled data (since, if I understand correctly, the success of the method does depend on prior model alignment) and (2) why not try with more examples than 250?

Line 448: It seems odd to me that ACT outperforms Zephyr SFT on the overall trajectory outcome but not action recognition. Does the A model show bias somehow to prefer asking clarifying questions?

Line 527: this paragraph can just be a single sentence--just tell us prior alignment was helpful and how much, don't need to discuss more.

Finally, there wasn't much (any?) discussion of the limitations of user simulations. In particular, I always worry that humans are less predictable than simulations--how might this affect your results?

**Details Of Ethics Concerns:**

Figure 3 and related text: This figure was critical to my understanding your approach. It would help to label the figure with the notation you use in the text, and refer to the specific scenarios labeled in the figure in the text as well. It took me some time to understand the text and this cross-labeling would have helped.

---

> ### Author Response · Authors · 2024-11-22
> **Response to Reviewer Qb6Q (1/2)**
>
> Thank you for your careful review and helpful feedback! We appreciate you acknowledging the soundness of our work with extensive experiments, the impact of our demonstration of the limitations of off-policy learning, and the novelty of our text-to-SQL dataset. We are grateful for your helpful suggestions which has greatly helped to strengthen our paper, and we have responded to each of them below.
>
> > **”The paper is…”**
>
> > **”probably over-stuffed with results”**
>
> > **”generally very wordy making it difficult to read. It is saved by the clear algorithms and figures, but the text should be shortened considerably. This would also allow for more discussion of the AmbigSQL data set, which is a significant contribution”**
>
> > **”You could also omit most or all of the DPO-Dist results from the main text--it never gets close to the best results”**
>
> > **”L 527: this paragraph can just be a single sentence--just tell us prior alignment was helpful and how much, don't need to discuss more.”**
>
> R: We appreciate your very helpful suggestions! We have condensed the writing in the paper, and we have reorganized our results, as follows:
>
> 1. Our supplementary baselines (e.g. DPO-Dist, prompting variants) have been moved to our Additional Experiments Section (Appendix Tables A7-9).
>
> 2. We have shortened our discussion of the ablations.
>
> 3. We have shortened the description of Implicit Ambiguity Recognition in Sec 4.2.
>
> 4. We have shortened the description of Agent task performance in Sec 4.2
>
> 5. We have merged our initial Qualitative Analysis section with the extended section in the Appendix (see Appendix Section I).
>
> 6. We have added a discussion on user simulators and action classifiers to the main text.
>
> 7. We have added more details on AmbigSQL in Sec 4.1.3 of the revised manuscript.
>
> We believe your suggestions have greatly helped to improve the presentation of our paper.
>
> > **”The AmbigSQL data might be too easy to learn, as some models achieve near perfect performance on it.”**
>
> R: It is true that with sufficient fine-tuning data, models are able to achieve near perfect performance on the implicit ambiguity recognition task on AmbigSQL. This is somewhat expected due to the three systematic perturbation patterns we detail in Appendix B.1. However, we believe that the value in AmbigSQL is the complexity of the multi-turn text-to-SQL task. Table A11 demonstrates that it is not possible to achieve high performance on the task without asking clarifying questions, and despite the promise of using tuning to learn the question-asking pattern in the data, Table 3 shows that Gemini struggles to learn question-asking using only limited in-context examples. Smaller models also struggle greatly on the SQL generation task.
>
> > **”One thing you could do after shortening the rest of the paper is to consider the use of user simulators more, and find parallels between earlier work and yours. I think the novelty of the paper lies mostly in the exact implementation of data generation for preference optimization, summarized in Algorithm 1 and Figure 3, and it would be good to focus on that specifically.”**
>
> > **”I would like to see details of the user simulator U and action classifier A in the main text.”**
>
> > **”there wasn't much discussion of the limitations of user simulations.”**
>
> R: We appreciate your suggestions on simulation. We have clarified the description of our approach and added the following discussion of user simulators to Section 3.1:
>
> “Works such as Deng et al. (2023c); Yu et al. (2023) directly prompt LLMs for goal-oriented tasks conditioned on dialogue context and task objectives.  Our implementation ofUis inspired by this setup. We first prompt an LLM to summarize the user’s information-seeking goal. Then, we form another prompt using this summary along with the current dialogue context to simulate a user response. Prompting with this goal summary allows for more flexibility than directly providing the user simulator with the ground truth information objective. We provide details on our implementation of U in Appendix H.”
>
> We have also added a discussion on Action Classifiers in the same section.
>
> > **“Figure 3 was critical to my understanding of your approach. It would help the text to label figure 3 with the notation used in the text”**
>
> R: Thanks for your helpful suggestion! Our revised paper contains an updated Figure 3 with the notation (temporarily written in red) in Section 3.1. Additionally, we have updated the iterator notation used in Algorithm 2 and Section 3.2.2 for clarity and consistency with the Figure. We hope that our changes help to create a balance between including a detailed explanation without overwhelming readers with too many details.

---

> ### Author Response · Authors · 2024-11-22
> **Response to Reviewer Qb6Q (2/2)**
>
> > **”I found the sections on "conversations in the wild" difficult to follow and disconnected from the rest of the text. … what specific problem you are trying to address?”**
>
> > **”L425-427: ACT does better than frontier LLMs when it has 250 examples to learn from. I assume you don't need human labeling.”**
>
> R: To clarify, ACT is a data-efficient tuning framework, and our main experiments use ground-truth action labels, which is commonly the case with academic datasets. Each dataset has dialogue turns which are labeled with whether they are ambiguous, which is the basis of “optimal”/“sub-optimal” actions. Our in-the-wild experiments refer to scenarios where these labels are not directly available. In Sec5.4 we have discussed this point, and Tab4 demonstrates our methods are effective in scenarios without label supervision.
>
> > **”if I understand correctly, the success of the method does depend on prior model alignment)”**
>
> R: The method does not rely on prior alignment, but intuitively, prior alignment provides a stronger initialization.
>
> > **“why not try with more examples than 250?”**
>
> R: For fixed tasks where you may expect your training data to cover the target distribution, it is possible to achieve strong performance with standard SFT. [1] finds most of the performance gains come from SFT and there is limited improvement even from on-policy DPO, and [2] finds SFT on high quality data can outperform RLAIF. [3] finds that SFT with hindsight relabeling can outperform many offline RL techniques. The exploration of on-policy tuning strategies like ACT shine especially in the setting where there is insufficient training data to cover the target task. Given these previous findings, we decided to explore customized tuning approaches for the low-data region.
>
> [1] LIONs: An Empirically Optimized Approach to Align Language Models, EMNLP 2024
>
> [2] A Critical Evaluation of AI Feedback for Aligning Large Language Models, NeurIPS 2024
>
> [3] RvS: What is Essential for Offline RL via Supervised Learning? At ICLR 2022
>
> > **”Algorithm 2 you say (line 14-16) that the trajectory is assigned to the winning or losing labels--but in the text it is just the sampled y_j that is assigned to these”**
>
> R: Thank you for pointing out this typo. As in Algorithm 2, y_{lj} is set to the incorrect trajectory. The trajectory contains the responses from the original sampled response y_j leading to the incorrect outcome g_j’. We have fixed this in the revised manuscript (L291-294).
>
> > **”Line 304: How do you guarantee that the responses from Gemini Ultra model M should be rejected? Won't it often come up with a good response?”**
>
> R: In Alg1, L3 says that we infer a ground truth pragmatic action (a_i), L4 states that we determine a conjugate rejected action (a_i’), and in L6 we prompt an LLM *conditioned on the rejected action (a_i’).*
>
> > **”Line 354: Why not stick with F1, even though AmbigSQL is balanced?”**
>
> R: In our updated experiments (which include an Iterative Reasoning Preference Optimization baseline), we have added Macro F1.
>
> | Condition | Tuning Conversations | Accuracy | Action F1 | Turn F1 | Trajectory F1 |
> |---------------|---------------|---------------|---------------|---------------|---------------|
> | Gemini | 10-shot | 72.9 | 70.9 | 63.5 | 75.2 |
> | Claude | 10-shot | 68.5 | 63.8 | 66.5 | 72.4 |
> | Zephyr SFT | 50 | 77.4 | 77.4 | 21.9 | 13.9 |
> | Zephyr IRPO | 50 | 91.0 | 91.0 | 27.8 | 30.8 |
> | Zephyr ACT | 50 | 80.8 | 80.7 | 43.6 | 38.1 |
> | Zephyr SFT | 100 | 97.2 | 97.2 | 43.3 | 34.3 |
> | Zephyr IRPO | 100 | 96.2 | 96.1 | 45.0 | 37.0 |
> | Zephyr ACT | 100 | 99.2 | 99.3 | 48.0 | 49.6 |
> | Zephyr SFT | 250 | 99.8 | 99.7 | 51.0 | 50.7 |
> | Zephyr IRPO | 250 | 97.0 | 97.1 | 49.7 | 45.6 |
> | Zephyr ACT | 250 | 99.9 | 99.8 | 52.3 | 53.0 |
>
> > **”Line 448: It seems odd to me ACT outperforms Zephyr SFT on overall trajectory outcome but not action recognition. Does the A model show bias to asking questions?”**
>
> R: The Action Classifier does not bias towards clarifying questions. It classifies an utterance to compare it against a ground-truth label rather than predicting one. In Tab 1-3, ACT does outperform Zephyr SFT on action recognition in nearly every condition. Even when SFT outperforms ACT in terms of action recognition (in Table 2), ACT outperforms SFT in multi-turn performance. This indicates that *ACT is able to better teach models how to reason about its own clarifying questions.*
>
> In Appendix A.1 of our revised paper, we have now noted that action recognition performance can be impacted by label noise (Abg-CoQA in particular has relatively low inter-annotator agreement on ambiguity compared to PACIFIC). Trajectory outcome performance is less dependent on labels and is improved by online sampling and multi-turn optimization. Our results indicate ACT is consistently able to improve upon baselines on multi-turn goal completion.
>
> We thank you again for your very thoughtful review and helpful feedback!

---

> > ### Author Response · Authors · 2024-11-24
> >
> > Dear Reviewer Qb6Q,
> >
> > We thank you again for your detailed review and your insightful comments! Your feedback has helped us to greatly refine the presentation of our paper, and your comments have been invaluable towards improving the overall framing and positioning of the novelty and significance of our work.
> >
> > We respectfully ask whether you may have any remaining concerns regarding our work, and we will do our best to address them.
> >
> > Kind regards,
> >
> > The Authors

---

> > > ### Comment · Reviewer_Qb6Q · 2024-11-25
> > >
> > > Updated ratings after reading author responses. In particular the presentation has improved and the paper is easier to understand. Thank you to the authors for their efforts.

---

> > > > ### Author Response · Authors · 2024-11-25
> > > >
> > > > Dear Reviewer Qb6Q,
> > > >
> > > > Thank you for acknowledging our response and increasing your ratings! We are very grateful for your insightful feedback which has helped to greatly strengthen our paper, and hope we have addressed all of your concerns.
> > > >
> > > > Best regards,
> > > >
> > > > The Authors

---

### Official Review · Reviewer_g3nF · 2024-11-04

**Soundness:** 3
**Presentation:** 3
**Contribution:** 3
**Rating:** 6
**Confidence:** 5

**Summary:**

The paper introduces Action-Based Contrastive Self-Training (ACT), an algorithm designed to enhance large language models' (LLMs) ability to handle ambiguity in multi-turn conversations. Current LLM-based conversational agents often fail to ask clarifying questions when faced with ambiguous input, opting instead to overhedge or guess the user's intent. ACT addresses this by enabling data-efficient dialogue policy learning through contrasting different conversational actions, even without explicit action labels, using a preference optimization approach based on Direct Preference Optimization (DPO).

The authors demonstrate ACT's effectiveness across several real-world tasks, including tabular-grounded question answering, machine reading comprehension, and AmbigSQL—a new task focused on disambiguating complex SQL queries. Their experiments show that ACT outperforms standard tuning methods like supervised fine-tuning and DPO, particularly in scenarios with limited data. Additionally, the paper proposes a workflow for evaluating LLMs' ability to recognize and reason about ambiguity in conversations. The results suggest that incorporating action-based preferences via ACT enhances LLMs' performance in modeling multi-turn dialogues and improves their conversational skills.

**Strengths:**

1. This paper is clear in writing. It is easy to understand the proposals and experiments.
2. Diversified tasks of the evaluation datasets. I recognize the efforts author made to experiment on different tasks (e.g. SQL, tableQA etc.)

**Weaknesses:**

1. The biggest weakness of the paper should be the choice of weak baselines and this japardizes the contributions claims of the paper.
One paper on the top of my head is https://arxiv.org/pdf/2404.19733. This paper has many identical settings: online sampling and a heuristic to filter trajectories. I believe this paper should be a baseline to compare.
Maybe the authors have similar ablation comparisons in Table 6. I would like to see some clear discussions on the baselines and ablations.


2. Related to weakness 1, this paper doesn't discuss DPO variants built for tasks that require multi-step trajectories. This paper is proposing a new DPO algorithm but it has a very small section to discuss relevant DPO works. I would suggest the authors check into DPOs for conversational tasks or reasoning tasks that require COT. This will provide a more justified view of the paper contributions.

**Questions:**

see above

---

> ### Author Response · Authors · 2024-11-22
> **Response to Reviewer g3nF (1/2)**
>
> Thank you for taking the time to carefully read our paper and provide helpful feedback. We appreciate your acknowledgement of the quality of our writing, as well as the extensiveness of our experiments across multiple domains. Below, we have responded to each of your helpful suggestions, which we believe has greatly helped to increase the quality of our paper.
>
> > *“The biggest weakness of the paper should be the choice of weak baselines … One paper is https://arxiv.org/pdf/2404.19733. This paper has many identical settings: online sampling and a heuristic to filter trajectories. … the authors have similar ablation comparisons in Table 6. I would like to see some clear discussions”*
>
> R: Thank you for your helpful feedback! You are right that there are some other concurrent works which have introduced on-policy learning into offline DPO-type methods recently, including IRPO (which was recently accepted at NeurIPS). ACT fundamentally differs from existing on-policy DPO variants (e.g., IRPO) in two aspects.
>
> First, ACT is designed for multi-turn *conversation modeling* specifically. While ACT includes online sampling combined with heuristic-based automatic evaluation, we believe our paper is one of the first works to consider multi-turn trajectories for preference optimization, and the first paper to consider contrastive learning on the basis of conversational actions. As you pointed out, we have ablated many of the components of ACT in Table 6. The ablation condition where we take away multi-turn simulation can be considered a special case of Online or Iterative DPO in which we have a special reward signal based on conversation actions. Our ablations found that perturbing the action signal greatly hurts performance across all metrics, and that removing multi-turn simulation greatly hurts clarification reasoning ability.
>
> Second, recent works such as IRPO have considered improving reasoning by self-improving explicit reasoning by optimizing reasoning chains. Our problem setup (Sec4) aims to internalize the reasoning process through direct preference optimization. The current formulation of ACT bypasses explicit reasoning, attempting to encourage the model to generate action-optimal responses directly. Explicit reasoning optimization approaches like IRPO follow the recently popular literature on optimizing strategies which leverage inference-time computation. ACT is compatible with such reasoning approaches, but this was not the focus of our work.
>
> We have included the above discussion in Section 2.2. Following your suggestion, we have also run new experiments comparing ACT against IRPO on PACIFIC and AmbigSQL, as IRPO has been demonstrated to achieve strong improvements in numerical reasoning tasks like GSM-8K. As a reward signal, we use the downstream task metrics (e.g. DROP F1) for each of our tasks.
>
> We find the following results on PACIFIC:
> | Condition | Tuning Conversations | Action F1 | Turn F1 | Trajectory F1 | Post-Clarify F1 |
> |---------------|---------------|---------------|---------------|---------------|---------------|
> | Zephyr SFT | 50 | 69.0 | 57.8 | 61.3 | 43.5 |
> | Zephyr IRPO | 50 | 67.7 | 59.1 | 56.7 | 34.4 |
> | Zephyr ACT | 50 | 82.2 (+21%) | 62.8 (+6%) | 61.9 (+9%) | 57.2 (+66%) |
> | Zephyr SFT | 100 | 82.3 | 58.6 | 60.3 | 49.9 |
> | Zephyr IRPO | 100 | 84.5 | 60.4 | 55.2 | 38.2 |
> | Zephyr ACT | 100 | 86.0  (+2%)| 65.0 (+8%) | 62.0 (+12%) | 57.4 (+50%) |
> | Zephyr SFT | 250 | 86.9 | 65.1 | 63.3 | 56.7 |
> | Zephyr IRPO | 250 | 85.4 | 64.9 | 58.4 | 40.3 |
> | Zephyr ACT | 250 | 89.6 (+5%) | 68.1(+5%) | 65.7 (+13%) | 62.0 (+54%) |
>
> We also obtain the following results on AmbigSQL:
> | Condition | Tuning Conversations | Action F1 | Turn F1 | Trajectory F1 |
> |---------------|---------------|---------------|---------------|---------------|
> | Zephyr SFT | 50 | 77.4 | 21.9 | 13.9 |
> | Zephyr IRPO | 50 | 91.0 | 27.8 | 30.8 |
> | Zephyr ACT | 50 | 80.8 (-11%) | 43.6 (+57%) | 38.1 (+24%) |
> | Zephyr SFT | 100 | 97.2 | 43.3 | 34.3 |
> | Zephyr IRPO | 100 | 96.2 | 45.0 | 37.0 |
> | Zephyr ACT | 100 | 99.2 (+3%) | 48.0 (+7%) | 49.6 (+34%) |
> | Zephyr SFT | 250 | 99.8 | 51.0 | 50.7 |
> | Zephyr IRPO | 250 | 97.0 | 49.7 | 45.6 |
> | Zephyr ACT | 250 | 99.9 (+3%) | 52.3 (+5%) | 53.0 (+16%) |
>
> We find that ACT outperforms IRPO tuning holistically, *even though IRPO optimizes the use of extra inference-time compute*. These findings are consistent with our Ablation studies in Table 5, which demonstrate that *action-based optimization and multi-turn trajectories are crucial to conversational modeling.* We have updated these results in the text in Tab. 1-3, and we have strengthened this discussion in Section 5.

---

> ### Author Response · Authors · 2024-11-22
> **Response to Reviewer g3nF (2/2)**
>
> > *“this paper doesn't discuss DPO variants built for tasks that require multi-step trajectories. … I would suggest the authors check into DPOs for conversational tasks or reasoning tasks that require COT. “*
>
> R: Thank you very much for your suggestion. At the time of submission, we cited several recent works examining on-policy variants of DPO, and following your feedback and we have now added new citations to the following concurrent works:
>
> [1] Iterative Reasoning Preference Optimization (NeurIPS 2024)
>
> [2] Parrot: Enhancing multi-turn instruction following for large language models (ACL 2024)
>
> [3] Self-Play Preference Optimization for Language Model Alignment (arXiv May 2024)
>
> [4] Self-Play Fine-Tuning Converts Weak Language Models to Strong Language Models (ICML 2024)
>
>
> Our related works section now includes this additional paragraph in Section 2.2:
> “Many of our contemporaries also question the limits of fully offline preference learning and have examined “online” variants of them (Guo et al., 2024; Xu et al., 2023; 2024b). Yuan et al. (2024) proposes iterative DPO, and Chen et al. (2024) proposes a variant where ground-truth responses are considered winning, and responses sampled from the previous iteration of a policy model are considered losing.  Pang et al. (2024) applies a variant of iterative DPO to optimize externalized reasoning chains. Our work differs from these in that we are proposing a novel approach to customize LLMs for specific conversational settings. While other works look at customizing preference data for DPO for conversations in general (e.g. Sun et al. (2024), the focus is still on single-turn response optimization. ACT considers multi-turn trajectories for preference optimization, and the first paper to consider contrastive learning on the basis of conversational actions.”
>
> If you have any suggestions for other more related conversational DPO works to discuss in our paper we would be glad to reference them!
>
> We really appreciate your helpful feedback and we hope that we have addressed each of your concerns.

---

> > ### Author Response · Authors · 2024-11-24
> >
> > Dear Reviewer g3nF,
> >
> > We thank you again for your insightful feedback! Your helpful comments have led us to greatly strengthen the baselines in our paper and improve the discussion of the differences between our tuning algorithm designed for conversations compared to  existing approaches which combine DPO with on-policy learning.
> >
> > We respectfully ask whether you may have any remaining concerns preventing any appropriate score adjustments, and we will do our best to address them.
> >
> > Thank you again for your time!
> >
> > Kind regards,
> >
> > The Authors

---

> > > ### Comment · Reviewer_g3nF · 2024-11-24
> > > **Response to Author Rebuttal**
> > >
> > > Thank you for addressing my comments in such a short time.
> > > The comparison experiments between IRPO and your framework do show that your framework has a clear advantage in conversational tasks and sequentially makes the paper stronger.
> > >
> > > I also see you have discussed closely related work and I believe that also makes the paper more complete.
> > > I am happy to raise my scores! I don't have any other questions.

---

> > > > ### Author Response · Authors · 2024-11-25
> > > >
> > > > Dear Reviewer g3nF,
> > > >
> > > > Thank you for acknowledging our response and increasing your scores! We are grateful for your valuable feedback and discussion, and hope we have sufficiently addressed all of your concerns.
> > > >
> > > > Best regards,
> > > >
> > > > The Authors

---

### Official Review · Reviewer_GK1A · 2024-11-12

**Soundness:** 3
**Presentation:** 3
**Contribution:** 3
**Rating:** 6
**Confidence:** 2

**Summary:**

This paper proposes a novel online optimization algorithm: action-based contrastive self-training, which facilitates data-efficient dialogue policy learning for modeling multi-turn conversation. The authors demonstrate their method on multiple real-world conversational tasks: tabular-grounded QA, machine reading comprehension and AmbigSQL. They also propose a way to evaluate LLM's ability to function as conversational agents. Their method ACT achieves substantial improvements than DPO and other baselines.

**Strengths:**

1. This paper proposes a feasible way to optimizing multi-turn conversation modeling for LLMs, which overcome some obstacles faced by existing methods.
2. Their proposed method ACT is sample-efficient based on online DPO, and achieves good performance on three real-world conversational tasks.
3. The paper is well-written and the experiments are convincing.

**Weaknesses:**

This paper conducts experiments on limited applications of dialogue tasks; there are also many variants for DPO based methods, however, the authors have not compared them as baselines.

**Questions:**

None

---

> ### Author Response · Authors · 2024-11-22
> **Response to Reviewer GK1A**
>
> Thank you for taking the time to review our paper, and for acknowledging the advantages of our multi-turn approach, our strong experimental results on a breadth of real-world tasks, and the clarity of our paper! We have addressed the two concerns you mentioned below.
>
> > **”experiments on limited applications of dialogue tasks”**
>
> R: We would like to mention that dialogue tasks are a broad and useful domain which have received a lot of attention in the last ten years. We would also like to clarify that we do not only focus on a single sub-domain of dialogue systems. Table 1 demonstrates our experimental results on PACIFIC, which involves reasoning about structured tables as well as natural language paragraphs describing various financial information. Table 2 demonstrates our experimental results on Abg-CoQA, which only reasons about natural language documents. Table 3 provides results on AmbigSQL, which is our ambiguous text-to-SQL task. Overall, our proposed method is task agnostic, and it is designed to improve downstream LLMs in task-specific ways for multi-turn conversational modeling.
>
> Although our focus is on ambiguous conversations, we also ran limited experiments on another planning task in a general goal-oriented interaction corpus [1] to demonstrate that our approach can also generalize to settings with non-conversational actions like tool use and a non-binary action space. The results below highlight that *our method achieves a higher average reward even on this action optimization task that does not contain explicit clarification labels in the data*.
>
> | Condition | Normalized Reward |
> |---------------|---------------|
> | Zephyr SFT | 0.538 |
> | Zephyr ACT | 0.592 |
>
> [1] Decision-oriented dialogue for human-ai collaboration, TACL 2024 https://arxiv.org/abs/2305.20076
>
> > **”there are also many variants for DPO based methods, however, the authors have not compared them as baselines.”**
>
> In our paper, we had originally considered a baseline in which we distill information from different LLMs using DPO, and we have provided many different experimental comparisons in the Additional Experiments in Appendix A.
>
> Following Reviewer g3nF’s suggestion, we have now additionally added comparisons to Iterative Reasoning Preference Optimization, a concurrent work which was recently accepted at NeurIPS 2024 and has received much attention. We find that our work greatly outperforms IRPO by up to 21% in terms of implicit action recognition and 12% in terms of trajectory-level F1. We have pasted those results here as well:
>
> PACIFIC:
>
> | Condition | Tuning Conversations | Action F1 | Turn F1 | Trajectory F1 | Post-Clarify F1 |
> |---------------|---------------|---------------|---------------|---------------|---------------|
> | Zephyr SFT | 50 | 69.0 | 57.8 | 61.3 | 43.5 |
> | Zephyr IRPO | 50 | 67.7 | 59.1 | 56.7 | 34.4 |
> | Zephyr ACT | 50 | 82.2 (+21%) | 62.8 (+6%) | 61.9 (+9%) | 57.2 (+66%) |
> | Zephyr SFT | 100 | 82.3 | 58.6 | 60.3 | 49.9 |
> | Zephyr IRPO | 100 | 84.5 | 60.4 | 55.2 | 38.2 |
> | Zephyr ACT | 100 | 86.0  (+2%)| 65.0 (+8%) | 62.0 (+12%) | 57.4 (+50%) |
> | Zephyr SFT | 250 | 86.9 | 65.1 | 63.3 | 56.7 |
> | Zephyr IRPO | 250 | 85.4 | 64.9 | 58.4 | 40.3 |
> | Zephyr ACT | 250 | 89.6 (+5%) | 68.1(+5%) | 65.7 (+13%) | 62.0 (+54%) |
>
> AmbigSQL:
> | Condition | Tuning Conversations | Action F1 | Turn F1 | Trajectory F1 |
> |---------------|---------------|---------------|---------------|---------------|
> | Zephyr SFT | 50 | 77.4 | 21.9 | 13.9 |
> | Zephyr IRPO | 50 | 91.0 | 27.8 | 30.8 |
> | Zephyr ACT | 50 | 80.8 (-11%) | 43.6 (+57%) | 38.1 (+24%) |
> | Zephyr SFT | 100 | 97.2 | 43.3 | 34.3 |
> | Zephyr IRPO | 100 | 96.2 | 45.0 | 37.0 |
> | Zephyr ACT | 100 | 99.2 (+3%) | 48.0 (+7%) | 49.6 (+34%) |
> | Zephyr SFT | 250 | 99.8 | 51.0 | 50.7 |
> | Zephyr IRPO | 250 | 97.0 | 49.7 | 45.6 |
> | Zephyr ACT | 250 | 99.9 (+3%) | 52.3 (+5%) | 53.0 (+16%) |
>
> We see that ACT outperforms IRPO tuning holistically, *even though IRPO optimizes the use of extra inference-time computation*. These findings are consistent with our Ablation studies in Table 5, which demonstrate that action-based optimization and multi-turn trajectories are crucial to conversational modeling.
>
> Thank you again for your helpful feedback, and we hope that we have addressed each of your concerns!

---

> > ### Author Response · Authors · 2024-11-24
> >
> > Dear Reviewer GK1A,
> >
> > We thank you again for your insightful comments! Your review has helped us to clarify that we have demonstrated our approach can be applied to impactful conversational domains ranging from machine reading comprehension to tabular QA to text-to-SQL generation. We have also managed to greatly strengthen our baselines and demonstrate the differences between our work and existing on-policy versions of DPO.
> >
> > We respectfully ask whether you may have any remaining concerns preventing any appropriate score adjustments, and we will do our best to address them.
> >
> > Thank you again for your time!
> >
> > Kind regards,
> >
> > The Authors

---

> > > ### Author Response · Authors · 2024-11-26
> > >
> > > Dear Reviewer GK1A,
> > >
> > > Thanks again for your helpful feedback! Your comments have helped us to better clarify the significance and novelties of our work as well as strengthen the experimental evidence provided to back our claims.
> > >
> > > As the discussion period comes to a close, we respectfully ask whether you may have any remaining questions or concerns that would prevent any appropriate score adjustments, and we will do our best to address them in the remaining time.
> > >
> > > Thank you again for your time and thoughtful review!
> > >
> > > Kind regards,
> > >
> > > The Authors

---

### Author Response · Authors · 2024-11-22
**Response to All Reviewers**

We would like to extend our sincere gratitude to the reviewers for their insightful feedback on  our paper. We are glad to see all reviewers recognize the value of this work and unique contribution of our work from the perspectives of clarifying question optimization and exploring the limits of off-policy learning. The helpful suggestions we have received have helped us to greatly strengthen our paper. To summarize, we have run the following experiments and made the appropriate changes to our manuscript on OpenReview:

---

1. Following Reviewer g3nF and Reviewer GK1A’s suggestion, we have run additional experiments comparing ACT to Iterative Reasoning Preference Optimization on our quantitative reasoning tasks, finding that ACT consistently outperforms IRPO in multi-turn evaluations. We have updated Table 1 and 3 in the manuscript with these results.

2. Following Reviewer Qb6Q’s suggestion, our other baselines (e.g. prompting variants) have been moved to our Additional Experiments Section (Appendix Tables A7-9).

3. Following Reviewer g3nF’s suggestion, we have added an additional description on the differences between ACT and existing on-policy variants of DPO which are not focused on conversational modeling improvements.

*Addressing the above points has helped us clarify the differences between our work and existing works which propose generic approaches to on-policy DPO.*

4. Following Reviewer thTU and Reviewer GK1A’s feedback, we have clarified that we do not only focus on a single dialogue domain. We have also run an additional experiment to demonstrate that ACT can generalize beyond a binary action space, to non-conversational actions, even though this is not the focus of our paper.

*This has helped us to reiterate that our work is not specific to SQL-based dialogue tasks, and demonstrated that ACT has the potential to generalize beyond optimizing the clarification action.*

5. Following Reviewer thTU’s feedback, we have demonstrated that even larger LLMs do not currently excel at multi-turn disambiguation and that this ability can be greatly improved through specialized tuning.

6. Following Reviewer thTU’s feedback, we have clarified that our results do not focus on a single LLM. We have run prompting experiments with multiple LLMs and we have run comparisons for ACT tuning with different base models.

*This has helped us to highlight the significance of our work, which is a general tuning approach to adapt LLMs to complex conversational settings which may require skills such as disambiguation.*

7. Following Reviewer Qb6Q’s suggestion, we have added a discussion on user simulators to the main text in Sec 3.1.

8. Following Reviewer Qb6Q’s suggestion, we have shortened the description of Agent task performance in Sec 4.2

9. Following Reviewer Qb6Q’s suggestion, we have shortened the description of Implicit Ambiguity Recognition in Sec 4.2.

10. Following Reviewer Qb6Q’s suggestion, we have added more details on AmbigSQL in Sec 4.1.3 of the revised manuscript.

11. Following Reviewer Qb6Q’s suggestion, we have shortened our discussion of the ablations.

12. We have merged our initial Qualitative Analysis section with the extended section in the Appendix (see Appendix Section I).

13. Following Reviewer Qb6Q’s suggestion, we have updated the notation in Figure 3.

*Incorporating these helpful suggestions have greatly improved the clarity, quality, and positioning of our paper.*

---

We are also appreciative of the reviewers’ acknowledging the various strengths of our paper, particularly:

Reviewer g3nF and GK1A acknowledged the *breadth of our experiments on three diverse real-world conversational tasks.*

Reviewer GK1A and thTU acknowledged the *novelty and significance of our proposed method for multi-turn modeling*.

Reviewer Qb6Q acknowledged the *soundness* of our work, our *contribution to the growing literature exploring the tradeoffs of off-policy and on-policy learning*, and the *novelty* of our AmbigSQL dataset.

Reviewer thTU, GK1A, and g3nF acknowledged the *quality and clarity of our writing*.

---

In conclusion, we would like to express our appreciation to the reviewers for their invaluable insights which have been immensely helpful in refining our paper. We have responded in detail regarding these points to each review individually, and we are committed to incorporating each point of feedback in our revisions.

---

### Author Response · Authors · 2024-12-02
**Thank you to all reviewers!**

Dear Reviewers,

As the discussion period comes to a close today, we would again like to express our sincere gratitude to each of you for your valuable insights and feedback! The feedback and discussion has greatly helped to strengthen our experiments, paper framing, and overall completeness. If there are any remaining questions or concerns you may have about our submission, we are happy to do our best to address them in the remaining time.

Thank you again for your time and efforts!

Kind regards,

The Authors

---

### Meta-Review · Area_Chair_z7VM · 2024-12-18

**Metareview:**

This paper presents Action-Based Contrastive Self-Training (ACT), a framework for fine-tuning LLMs to address ambiguity in multi-turn conversations. The approach leverages a preference optimization objective, building on DPO, to contrast different conversational actions without requiring explicit action labels. ACT is demonstrated on tasks including tabular-grounded question answering, machine reading comprehension, and AmbigSQL, a new dataset designed for disambiguating SQL queries. The paper also introduces a workflow for evaluating LLMs' ability to handle ambiguity in dialogues.

The strengths of the paper are:

1. The paper introduces a novel online DPO method, ACT, which demonstrates sample efficiency and effectively addresses challenges in multi-turn conversational modeling for LLMs.
2. The authors provide experimental results across various conversational tasks, including SQL and TableQA, showing some performance improvements, e.g., in some cases 10–20% over existing methods like SFT and modified DPO.
3. The inclusion of the new AmbigSQL dataset could be a valuable resource to the field, enabling more straightforward automated evaluation and furthering research on handling ambiguity in conversational tasks.

As the paper has been extensively extended during the author response period, I will focus on the concerns that seem to remain in the current version:

1. While the authors provided clarifications on their DPO method and highlighted its focus on conversational modeling improvements, the method of the paper doesn't seem very creative or original. It is yet another variant of DPO aimed at a particular goal (clarifying question optimization).

 2. Experimental results: The most compelling results seem to be on the PACIFIC dataset, but this dataset is small and domain-specific (finance). The models evaluated in this work are quite small scale, not going beyond 7B. The experiments shown in the paper rely on 250 conversations or less, but without data scaling curves, it is more difficult to appreciate the contribution of the proposed method. As sample training size is small for all the experiments, it would be important to show mean and variance across different runs.

During the reviewer–AC discussion period, most reviewers sided with accepting the paper, but the dissenting reviewer kept their rating at 5 and expressed concerns about the experiments (see weakness 2, which is mostly a paraphrase of their points). I think this reviewer makes valid points, and the authors should seriously consider their concerns for the next version of the paper. However, I think these concerns are not serious enough to override the other reviewers' consensus to accept, so I recommend acceptance.

**Additional Comments On Reviewer Discussion:**

The authors extensively modified the paper during the discussion period, and the changes would be too many to list here (See response to all reviewers). The reviewers were mostly satisfied with the changes, but reviewer thTU remains concerned (see Weakness 2 above). For the camera-ready paper, I strongly recommend that the authors try to address these concerns.

---

### Decision · Program_Chairs · 2025-01-22

Accept (Poster)